# Normalization Helps Training of Quantized LSTM

**Lu Hou[1], Jinhua Zhu[2], James T. Kwok[1], Fei Gao[3], Tao Qin[3], Tie-yan Liu[3]**
[1]Hong Kong University of Science and Technology, Hong Kong
{lhouab,jamesk}@cse.ust.hk
[2]University of Science and Technology of China, Hefei, China
teslazhu@mail.ustc.edu.cn
[3]Microsoft Research, Beijing, China
{feiga, taoqin, tyliu}@microsoft.com

## Abstract

The long-short-term memory (LSTM), though powerful, is memory and computation expensive. To alleviate this problem, one approach is to compress its weights by quantization. However, existing quantization methods usually have inferior performance when used on LSTMs. In this paper, we first show theoretically that training a quantized LSTM is difficult because quantization makes the exploding gradient problem more severe, particularly when the LSTM weight matrices are large. We then show that the popularly used weight/layer/batch normalization schemes can help stabilize the gradient magnitude in training quantized LSTMs. Empirical results show that the normalized quantized LSTMs achieve significantly better results than their unnormalized counterparts. Their performance is also comparable with the full-precision LSTM, while being much smaller in size.

## 1 Introduction

The long-short-term memory (LSTM) [10] has achieved remarkable performance in various sequence modeling tasks [26, 28, 14]. Though powerful, the high-dimensional input/hidden/output and recursive computation across long time steps lead to space and time inefficiencies [29, 1], limiting its use on low-end devices with limited hardware resources.

In the LSTM, its weight matrices account for most of the time and space complexities. To lighten the computational demands, a popular approach is to quantize each weight to fewer bits. Previous weight quantization methods are mainly used on feedforward networks. In BinaryConnect [5], each weight is binarized. By introducing a scaling parameter, the binary weight network [23], ternary weight network [17], and loss-aware binarized/ternarized network [11, 12] often report performance that are even comparable with the full-precision network. However, when used to quantize LSTMs, their performance is usually inferior, and BinaryConnect even fails [11, 1]. To alleviate this problem, one has to use more bits and/or much more sophisticated quantization functions [9, 29, 22, 18].

On the other hand, normalization techniques, such as weight normalization [24], layer normalization [2] and batch normalization [13], have been found useful in improving deep network training and performance. In particular, while batch normalization is initially limited to feedforward networks, it has been recently extended to LSTMs [4]. Very recently, Ardakani et al. [1] used this extension to train binarized/ternarized LSTMs, and achieved state-of-the art performance. However, it remains unclear why batch normalization works well on quantized LSTMs, and also leaves open the question whether weight normalization and layer normalization may also work. Besides, the batch normalization extension in [4, 1] requires storing separate full-precision population statistics for each time step. These can cost even more storage than the quantized LSTM model itself. Moreover, it has to be used with a stochastic weight quantization function, which can be expensive due to the sampling operation.

In this paper, we first study theoretically why the quantized LSTM is difficult to train. Similar to the analysis on vanilla RNN in [21], we study the LSTM by investigating an upper bound on its backpropagated gradient w.r.t. the hidden states. We show that in each backpropagation step, the scale of this gradient is controlled by a number bounded by the norm of LSTM weights. Quantization tends to increase these norms, particularly for large models, making the exploding gradient problem much more severe than its full-precision counterpart. We then study the quantized LSTM with weight, layer, and batch normalization. Unlike the batch-normalized LSTM in [1] which requires a new stochastic weight quantization, we propose to apply normalization directly on top of any existing quantization method. We show that these normalization methods make the gradient invariant to weight scaling, and thus alleviate the problem of having a possibly large weight norm increase caused by quantization. Experiments are performed on various quantization methods with weight/layer/batch normalization. Results show that all three normalization schemes work well with quantized LSTMs, and achieve better results than their unnormalized counterparts. With only one or two bits, the normalized quantized LSTMs achieve comparable performance with the full-precision baseline. Moreover, weight/layer normalization perform as well as batch normalization (with separate statistics), but are more memory efficient.

**Notations:** For a vector $\mathbf{x}$, $\|\mathbf{x}\| = \sqrt{\sum_i x_i^2}$ is its $\ell_2$-norm, and $\mathrm{Diag}(\mathbf{x})$ returns a diagonal matrix with $\mathbf{x}$ on the diagonal. For two vectors $\mathbf{x}$ and $\mathbf{y}$, $\mathbf{x} \odot \mathbf{y}$ denotes the element-wise multiplication. For a matrix $\mathbf{X}$, $\|\mathbf{X}\|_2$ is its spectral norm (which is equal to its largest singular value).

## 2 Problem of Exploding Gradient

It is well-known that the vanilla RNN suffers from exploding and vanishing gradient problems due to long-term dependencies [3, 21]. To avoid vanishing gradient, the LSTM employs self-connection in the cell [10]. Its recurrence (without using peephole connections) is:

$$\begin{bmatrix} \mathbf{i}_t \\ \mathbf{f}_t \\ \mathbf{a}_t \\ \mathbf{o}_t \end{bmatrix} = \begin{bmatrix} \mathbf{W}_{xi}\mathbf{x}_t + \mathbf{W}_{hi}\mathbf{h}_{t-1} \\ \mathbf{W}_{xf}\mathbf{x}_t + \mathbf{W}_{hf}\mathbf{h}_{t-1} \\ \mathbf{W}_{xa}\mathbf{x}_t + \mathbf{W}_{ha}\mathbf{h}_{t-1} \\ \mathbf{W}_{xo}\mathbf{x}_t + \mathbf{W}_{ho}\mathbf{h}_{t-1} \end{bmatrix} + \begin{bmatrix} \mathbf{b}_i \\ \mathbf{b}_f \\ \mathbf{b}_a \\ \mathbf{b}_o \end{bmatrix}, \tag{1}$$

$$\mathbf{c}_t = \sigma(\mathbf{i}_t) \odot \tanh(\mathbf{a}_t) + \sigma(\mathbf{f}_t) \odot \mathbf{c}_{t-1}, \tag{2}$$

$$\mathbf{h}_t = \sigma(\mathbf{o}_t) \odot \tanh(\mathbf{c}_t). \tag{3}$$

Here, $\mathbf{x}_t, \mathbf{h}_t$ and $\mathbf{c}_t$ are the input, hidden state and cell state at time $t$, $\mathbf{W}_{xi}, \mathbf{W}_{xf}, \mathbf{W}_{xa}, \mathbf{W}_{xo} \in \mathbb{R}^{d \times r}$, $\mathbf{W}_{hi}, \mathbf{W}_{hf}, \mathbf{W}_{ha}, \mathbf{W}_{ho} \in \mathbb{R}^{d \times d}$ are the weight matrices, and $\mathbf{b}_i, \mathbf{b}_f, \mathbf{b}_a, \mathbf{b}_o \in \mathbb{R}^d$ are the biases.

In the following, we show that the gradients in the LSTM can still explode. For the vanilla RNN, the backpropagated gradient takes the form of a product of Jacobian matrices. By studying the upper bound on gradient magnitude, the necessary condition for exploding gradient can be derived [21]. In this paper, we follow the same approach, and study upper bounds on the gradient magnitude in the LSTM. Because of the introduction of $\mathbf{c}_t$, the backpropagated gradient in LSTM is not of the simple form as that for vanilla RNN, and the upper bound analysis is much more difficult.

Lower bounds are more desirable in deriving a sufficient condition for exploding gradient. However, even for the vanilla RNN, only an upper bound can be derived [21]. Moreover, as will be demonstrated empirically in Section 4, these upper bounds can still help explain the behavior of exploding gradient (Figures 2-3) and failure of BinaryConnect and TerConnect in quantized LSTM (Tables 2-4).

### 2.1 Exploding Gradient in LSTM

Let the loss be $\xi = \sum_{m=1}^{T} \xi_m$, where $T$ is the number of time steps unrolled, and $\xi_m$ is the loss at time step $m$. In backpropagation, recall that we first obtain $\frac{\partial \xi_m}{\partial \mathbf{h}_m}$ and $\frac{\partial \xi_m}{\partial \mathbf{c}_m}$, and then backpropagate from $\frac{\partial \xi_m}{\partial \mathbf{h}_t}$ and $\frac{\partial \xi_m}{\partial \mathbf{c}_t}$ to $\frac{\partial \xi_m}{\partial \mathbf{h}_{t-1}}$ and $\frac{\partial \xi_m}{\partial \mathbf{c}_{t-1}}$ (for $t \leq m$). We study the exploding gradient problem of LSTM by first considering $\|\frac{\partial \xi_m}{\partial \mathbf{h}_{t-1}}\|$ and $\|\frac{\partial \xi_m}{\partial \mathbf{h}_t}\|$ at adjacent time steps. Let $\gamma_1 = \max_{1 \leq t \leq m, 1 \leq j \leq d} |[\mathbf{c}_{t-1}]_j|$, and

$$\lambda_1 = \frac{1}{4}\|\mathbf{W}_{hi}\|_2 + \frac{\gamma_1}{4}\|\mathbf{W}_{hf}\|_2 + \|\mathbf{W}_{ha}\|_2 + \frac{1}{4}\|\mathbf{W}_{ho}\|_2, \quad \lambda_2 = \frac{1}{4}\|\mathbf{W}_{hi}\|_2 + \frac{\gamma_1}{4}\|\mathbf{W}_{hf}\|_2 + \|\mathbf{W}_{ha}\|_2. \tag{4}$$

**Proposition 2.1** $\|\frac{\partial \xi_m}{\partial \mathbf{h}_{t-1}}\| \leq \lambda_1 \|\frac{\partial \xi_m}{\partial \mathbf{h}_t}\| + \lambda_2 \|\frac{\partial \xi_m}{\partial \mathbf{c}_{t+1}}\|$.

When $\lambda_2 = 0$, Proposition 2.1 simplifies to $\|\frac{\partial \xi_m}{\partial \mathbf{h}_{t-1}}\| \leq \lambda_1 \|\frac{\partial \xi_m}{\partial \mathbf{h}_t}\|$. By induction, for any time step $p < t$, $\|\frac{\partial \xi_m}{\partial \mathbf{h}_p}\| \leq \lambda_1^{t-p} \|\frac{\partial \xi_m}{\partial \mathbf{h}_t}\|$. The norm of this backpropagated gradient can grow exponentially when $\lambda_1 > 1$, leading to exploding gradient. Hence, we have the following corollary.

**Corollary 1** *When $\lambda_2 = 0$, a necessary condition for exploding gradients in the LSTM is $\lambda_1 > 1$.*

Empirically, $\lambda_2$ is rarely zero (Figure 1). The upper bound of $\|\frac{\partial \xi_m}{\partial \mathbf{h}_{t-1}}\|$ in Proposition 2.1 is then even larger, and the gradient may explode even more easily.

## 2.2 Exploding Gradient in Quantized LSTM

From (1)-(3), most of the LSTM's parameters are due to matrices $\mathbf{W}_{xi}, \mathbf{W}_{xf}, \mathbf{W}_{xa}, \mathbf{W}_{xo}, \mathbf{W}_{hi}, \mathbf{W}_{hf}, \mathbf{W}_{ha}, \mathbf{W}_{ho}$. In the sequel, we use $\mathbf{W}_{x*}$ and $\mathbf{W}_{h*}$ to denote these matrices. The computation is also dominated by matrix-vector multiplications of the form $\mathbf{W}_{x*}\mathbf{x}_t + \mathbf{W}_{h*}\mathbf{h}_{t-1}$ [1]. Quantizing these weight matrices can thus significantly reduce space and time [11, 29, 1].

The following propositions show that a large $d$ leads to a large $\|\mathbf{W}_{h*}\|_2$ for both the binarized LSTM (Proposition 2.2) and $m$-bit quantized LSTM (Proposition 2.3).

**Proposition 2.2** *[11] For any $\mathbf{W} \in \{-1, +1\}^{d \times d}$, $\|\mathbf{W}\|_2 \geq \sqrt{d}$. Equality holds iff all singular values of $\mathbf{W}$ are the same.*

**Proposition 2.3** *For any $\mathbf{W} \in \{-B_k, \ldots, -B_1, B_0, B_1, \ldots, B_k\}^{d \times d}$ where $0 = B_0 < B_1 < \cdots < B_k$, we have $\|\mathbf{W}\|_2 \geq (1-s)B_1\sqrt{d}$, where $s$ is the sparsity (fraction of zero elements) in $\mathbf{W}$. Equality holds iff all singular values of $\mathbf{W}$ are the same.*

For ternarization, $s > 0$ and $B_1 = 1$, the lower bound in Proposition 2.3 is smaller than that for binarization in Proposition 2.2. Table 1 compares the spectral norms of $\mathbf{W}_{h*}$ before/after binarization (using BinaryConnect) and ternarization (using TerConnect in Section 4). As can be seen, quantization increases its spectral norm, especially for large $d$ and for binarization. When $\|\mathbf{W}_{h*}\|_2$ becomes larger after quantization, $\lambda_1, \lambda_2$ in (4) also become large, and the exploding gradient problem becomes more severe. Empirically, though BinaryConnect [5] achieves remarkable performance on feedforward networks, it fails on LSTMs [11].

Table 1: Average spectral norm of 10 $d \times d$ weight matrices (obtained by various initialization methods) before/after binarization and ternarization. "Pytorch default" refers to the default uniform initialization used in the PyTorch implementation of LSTM. For ternarization, sparsity of the matrix is around $0.35$ empirically.

| $d$ | full-precision | | | binarized | ternarized |
| --- | --- | --- | --- | --- | --- |
| | Pytorch default | Xavier initialization [6] | He initialization [8] | | |
| 512 | $1.15 \pm 0.01$ | $1.98 \pm 0.01$ | $2.81 \pm 0.02$ | $44.76 \pm 0.28$ | $36.18 \pm 0.18$ |
| 1024 | $1.15 \pm 0.00$ | $1.99 \pm 0.01$ | $2.81 \pm 0.01$ | $63.64 \pm 0.30$ | $51.33 \pm 0.28$ |
| 2048 | $1.15 \pm 0.00$ | $2.00 \pm 0.01$ | $2.82 \pm 0.01$ | $90.32 \pm 0.30$ | $72.81 \pm 0.17$ |

To alleviate the exploding gradient problem, empirical success has been observed by adding a scaling factor to the quantized weight [11, 12]. For example, in binarization, the binarized values become $\{-\alpha, +\alpha\}$ for some $\alpha > 0$. We speculate that the success behind this simple method is that for any $\alpha \geq 0$, $\|\alpha \mathbf{W}_{h*}\|_2 = \alpha\|\mathbf{W}_{h*}\|_2$. By using $\alpha < 1$, the norm becomes smaller and the exploding gradient problem can be alleviated (empirical results are shown in Appendix B). However, $\alpha$ is usually learned and there is no guarantee that it is small enough to compensate for the increase in $\|\mathbf{W}_{h*}\|_2$ caused by quantization.

## 3 Normalization in LSTM

In this section, we theoretically study the properties of (full-precision and quantized) LSTMs with weight normalization [24], layer normalization [2], and batch normalization [13], and how these

properties help optimization of the quantized LSTMs. In general, let the normalization function be $\mathcal{N}(\cdot)$. The normalized LSTM satisfies the recurrence:

$$\begin{bmatrix} \tilde{\mathbf{i}}_t \\ \tilde{\mathbf{f}}_t \\ \tilde{\mathbf{a}}_t \\ \tilde{\mathbf{o}}_t \end{bmatrix} = \begin{bmatrix} \mathcal{N}(\mathbf{W}_{xi}\mathbf{x}_t) + \mathcal{N}(\mathbf{W}_{hi}\mathbf{h}_{t-1}) \\ \mathcal{N}(\mathbf{W}_{xf}\mathbf{x}_t) + \mathcal{N}(\mathbf{W}_{hf}\mathbf{h}_{t-1}) \\ \mathcal{N}(\mathbf{W}_{xa}\mathbf{x}_t) + \mathcal{N}(\mathbf{W}_{ha}\mathbf{h}_{t-1}) \\ \mathcal{N}(\mathbf{W}_{xo}\mathbf{x}_t) + \mathcal{N}(\mathbf{W}_{ho}\mathbf{h}_{t-1}) \end{bmatrix} + \begin{bmatrix} \mathbf{b}_i \\ \mathbf{b}_f \\ \mathbf{b}_a \\ \mathbf{b}_o \end{bmatrix}, \tag{5}$$

$$\mathbf{c}_t = \sigma(\tilde{\mathbf{i}}_t) \odot \tanh(\tilde{\mathbf{a}}_t) + \sigma(\tilde{\mathbf{f}}_t) \odot \mathbf{c}_{t-1}, \tag{6}$$

$$\mathbf{h}_t = \sigma(\tilde{\mathbf{o}}_t) \odot \tanh(\mathbf{c}_t). \tag{7}$$

## 3.1 Weight Normalization ($\mathcal{WN}$)

Weight normalization re-parameterizes the weight vector to decouple its length from direction. In a LSTM, each row $\mathbf{W}_{j,:}$ of a weight matrix $\mathbf{W}$ (where $\mathbf{W}$ can be $\mathbf{W}_{h*}$ or $\mathbf{W}_{x*}$) is separately normalized. The $j$th element of a weight-normalized vector $\mathcal{WN}(\mathbf{W}\mathbf{x})$ is $\mathcal{WN}(\mathbf{W}_{j,:}\mathbf{x}) = g_j \frac{\mathbf{W}_{j,:}}{\|\mathbf{W}_{j,:}\|}\mathbf{x}$, where $g_j$ is a trainable scaling factor. For $\mathbf{W}_{h*}$, let the corresponding $g_*$ be the largest $g_j$ across all rows of $\mathcal{WN}(\mathbf{W}_{h*}\mathbf{x})$, and $\mathbf{D}_* = \text{Diag}([\|(\mathbf{W}_{h*})_{1,:}\|, \|(\mathbf{W}_{h*})_{2,:}\|, \ldots, \|(\mathbf{W}_{h*})_{d,:}\|]^\top)$.

**Proposition 3.1** *With weight normalization,*

$$\left\| \frac{\partial \xi_m}{\partial \mathbf{h}_{t-1}} \right\| \leq \left( \frac{g_i}{4}\|\mathbf{D}_i^{-1}\mathbf{W}_{hi}\|_2 + \frac{\gamma_1 g_f}{4}\|\mathbf{D}_f^{-1}\mathbf{W}_{hf}\|_2 + g_a\|\mathbf{D}_a^{-1}\mathbf{W}_{ha}\|_2 + \frac{g_o}{4}\|\mathbf{D}_o^{-1}\mathbf{W}_{ho}\|_2 \right) \left\| \frac{\partial \xi_m}{\partial \mathbf{h}_t} \right\|$$

$$+ \left( \frac{g_i}{4}\|\mathbf{D}_i^{-1}\mathbf{W}_{hi}\|_2 + \frac{\gamma_1 g_f}{4}\left\|\mathbf{D}_f^{-1}\mathbf{W}_{hf}\right\|_2 + g_a\|\mathbf{D}_a^{-1}\mathbf{W}_{ha}\|_2 \right) \left\| \frac{\partial \xi_m}{\partial \mathbf{c}_{t+1}} \right\|.$$

Compared to the unnormalized LSTM (Proposition 2.1), the norm of the backpropagated gradient is now related to $g_*$'s and $\mathbf{D}_*$'s. As will be demonstrated in Appendix C, the $g_*$ value only increases slightly after quantization, and so we ignore its effect in the theoretical analysis. When $\mathbf{W}_{h*}$ is scaled by a factor $\alpha$, $\mathbf{D}_*$ will also be scaled by $\alpha$, and so $\mathbf{D}_*^{-1}\mathbf{W}_{h*}$ is not affected. Hence, the backpropagation of $\|\frac{\partial \xi_m}{\partial \mathbf{h}_t}\|$ in the quantized LSTM is not affected by the possibly large scaling of the weight matrix caused by quantization (Propositions 2.2 and 2.3), and the exploding gradient problem can be alleviated.

## 3.2 Layer Normalization ($\mathcal{LN}$)

Layer normalization normalizes the neuron activities in each layer to zero mean and unit variance, and can stabilize the hidden state dynamics for RNNs. Let $\mathbf{x} \in \mathbb{R}^d$ be the input (which can be $\mathbf{W}_{x*}\mathbf{x}_t$ or $\mathbf{W}_{h*}\mathbf{h}_{t-1}$ in (5)) to layer normalization, with mean $\mu$ and standard deviation $\sigma$ computed over its $d$ elements. Let $\mathbf{z} = (\mathbf{x} - \mu\mathbf{1})/\sigma$ be the z-normalized vector (with zero mean and unit variance). The output from layer normalization is $\mathbf{y} = \mathcal{LN}(\mathbf{x}) = \mathbf{g} \odot \mathbf{z} + \mathbf{b}$, where $\mathbf{g}$ and $\mathbf{b}$ are the scaling and bias parameters. For layer normalization applied to $\mathbf{W}_{h*}\mathbf{h}_{t-1}$, let $g_* = g_k, \sigma_* = \sigma_k$, where $k = \arg\max_{1 \leq j \leq d} g_j$.

**Proposition 3.2** *With layer normalization,*

$$\left\| \frac{\partial \xi_m}{\partial \mathbf{h}_{t-1}} \right\| \leq \left( \frac{1}{4}\frac{g_i}{\sigma_i}\|\mathbf{W}_{hi}\|_2 + \frac{\gamma_1}{4}\frac{g_f}{\sigma_f}\|\mathbf{W}_{hf}\|_2 + \frac{g_a}{\sigma_a}\|\mathbf{W}_{ha}\|_2 + \frac{1}{4}\frac{g_o}{\sigma_o}\|\mathbf{W}_{ho}\|_2 \right) \left\| \frac{\partial \xi_m}{\partial \mathbf{h}_t} \right\|$$

$$+ \left( \frac{1}{4}\frac{g_i}{\sigma_i}\|\mathbf{W}_{hi}\|_2 + \frac{\gamma_1}{4}\frac{g_f}{\sigma_f}\|\mathbf{W}_{hf}\|_2 + \frac{g_a}{\sigma_a}\|\mathbf{W}_{ha}\|_2 \right) \left\| \frac{\partial \xi_m}{\partial \mathbf{c}_{t+1}} \right\|.$$

If the elements of $\mathbf{W}_{h*}$ grow twice as large, the corresponding $\sigma_*$ will be twice as large, and $\|\mathbf{W}_{h*}\|_2/\sigma_*$ remains unchanged. Thus, the backpropagation of $\|\frac{\partial \xi_m}{\partial \mathbf{h}_t}\|$ is again not affected by scaling of $\mathbf{W}_{h*}$.

## 3.3 Batch Normalization ($\mathcal{BN}$)

Recently, batch normalization has achieved state-of-the-art performance with quantized LSTMs [1]. However, the underlying reason remains unclear. Besides, it has to be used with a stochastic weight

quantization function, which is expensive due to the underlying sampling operation. It is also memory expensive as separate mean and variance statistics for different time steps have to be stored.

In this work, we propose to directly apply batch normalization on top of any existing quantization method in LSTM. As will be seen in Section 4, this yields comparable or even better performance than [1], and is much cheaper in space when the batch statistics are shared across different time steps.

Batch normalization operates on a minibatch. At time $t$, let $\mathbf{x}_t^k \in \mathbb{R}^r$, $\mathbf{h}_t^k$, $\mathbf{c}_t^k \in \mathbb{R}^d$ be the input, hidden state and cell state for sample $k$ in a minibatch of $N$ samples, and $\mathbf{H}_t = [\mathbf{h}_t^1, \ldots, \mathbf{h}_t^N]^\top \in \mathbb{R}^{N \times d}$, $\mathbf{X}_t = [\mathbf{x}_t^1, \ldots, \mathbf{x}_t^N]^\top \in \mathbb{R}^{N \times r}$. The input to batch normalization is $\mathbf{X} \in \mathbb{R}^{N \times d}$ (which can be $\mathbf{X}_t \mathbf{W}_{x*}^\top$ or $\mathbf{H}_{t-1} \mathbf{W}_{h*}^\top$), with mean $\mu_j$ and standard deviation $\sigma_j$ for the $j$th column. The batch normalization output $\mathbf{Y} \in \mathbb{R}^{N \times d}$ has $\mathbf{Y}_{:,j} = \mathcal{BN}(\mathbf{X}_{:,j}) = g_j \frac{\mathbf{X}_{:,j} - \mu_j \mathbf{1}}{\sigma_j} + b_j \mathbf{1}$, where $\mathbf{g} = [g_1, \ldots, g_d]^\top \in \mathbb{R}^d$ and $\mathbf{b} = [b_1, \ldots, b_d]^\top \in \mathbb{R}^d$ are the scaling parameters and biases, respectively. For batch normalization applied to $\mathbf{H}_{t-1} \mathbf{W}_{h*}^\top$, let $(\sigma_*, g_*) = \arg\max_{\{\sigma_1, \ldots, \sigma_d\}, \{g_1, \ldots, g_d\}} \frac{g_j}{\sigma_j}$. Let $\gamma_2 = \max_{1 \le t \le m, 1 \le j \le d, 1 \le k \le N} |[\mathbf{c}_{t-1}^k]_j|$.

**Proposition 3.3** *For the unnormalized LSTM in (1),*

$$
\sum_{k=1}^N \left\| \frac{\partial \xi_m}{\partial \mathbf{h}_{t-1}^k} \right\|^2 \le \left( \frac{1}{2} \|\mathbf{W}_{hi}\|_2^2 + \frac{\gamma_2^2}{2} \|\mathbf{W}_{hf}\|_2^2 + 8 \|\mathbf{W}_{ha}\|_2^2 + \frac{1}{4} \|\mathbf{W}_{ho}\|_2^2 \right) \sum_{k=1}^N \left\| \frac{\partial \xi_m}{\partial \mathbf{h}_t^k} \right\|^2
$$
$$
+ \left( \frac{1}{2} \|\mathbf{W}_{hi}\|_2^2 + \frac{\gamma_2^2}{2} \|\mathbf{W}_{hf}\|_2^2 + 8 \|\mathbf{W}_{ha}\|_2^2 \right) \sum_{k=1}^N \left\| \frac{\partial \xi_m}{\partial \mathbf{c}_{t+1}^k} \right\|^2.
$$

**Proposition 3.4** *With batch normalization,*

$$
\sum_{k=1}^N \left\| \frac{\partial \xi_m}{\partial \mathbf{h}_{t-1}^k} \right\|^2 \le \left( \frac{1}{2} \frac{g_i^2}{\sigma_i^2} \|\mathbf{W}_{hi}\|_2^2 + \frac{\gamma_2^2}{2} \frac{g_f^2}{\sigma_f^2} \|\mathbf{W}_{hf}\|_2^2 + 8 \frac{g_a^2}{\sigma_a^2} \|\mathbf{W}_{ha}\|_2^2 + \frac{1}{4} \frac{g_o^2}{\sigma_o^2} \|\mathbf{W}_{ho}\|_2^2 \right) \sum_{k=1}^N \left\| \frac{\partial \xi_m}{\partial \mathbf{h}_t^k} \right\|^2
$$
$$
+ \left( \frac{1}{2} \frac{g_i^2}{\sigma_i^2} \|\mathbf{W}_{hi}\|_2^2 + \frac{\gamma_2^2}{2} \frac{g_f^2}{\sigma_f^2} \|\mathbf{W}_{hf}\|_2^2 + 8 \frac{g_a^2}{\sigma_a^2} \|\mathbf{W}_{ha}\|_2^2 \right) \sum_{k=1}^N \left\| \frac{\partial \xi_m}{\partial \mathbf{c}_{t+1}^k} \right\|^2.
$$

In contrast to the unnormalized LSTM (Proposition 3.3), when batch normalization is used, if the elements of $\mathbf{W}_{h*}$ in the quantized LSTM grow twice as large, the corresponding $\sigma_*$ will be twice as large, and $\|\mathbf{W}_{h*}\|_2^2 / \sigma_*^2$ remains unchanged. Thus, it is again unaffected by the scaling of $\mathbf{W}_{h*}$.

**Remark 3.1** *In summary, by using weight, layer or batch normalization, backpropagation of $\|\frac{\partial \xi_m}{\partial \mathbf{h}_t}\|$ in the quantized LSTM is not affected by the possibly large scaling of the weight matrix caused by quantization, and the exploding gradient problem can be alleviated.*

**Remark 3.2** *Note that the storage requirements of the normalization schemes differ. The full-precision LSTM requires $32 \times 4(rd + d^2 + d)$ bits to store the $\mathbf{W}_{x*}$'s, $\mathbf{W}_{h*}$'s, and $\mathbf{b}_*$'s in (1), while the $m$-bit unnormalized LSTM requires $m \times 4(rd + d^2) + 32 \times 4d$ bits. When normalization is used on the $m$-bit LSTM, weight normalization requires $32 \times 8d$ additional bits to store the scaling parameters $g_j$'s. Layer normalization is slightly more expensive, and needs $32 \times 16d$ bits to store the scaling parameters and biases. Batch normalization needs $32 \times 32d$ extra bits when the mean and variance statistics are shared across time steps, which is even more expensive but still small (compared to the LSTM size). However, when separate statistics are used in each time step, the additional space becomes $32 \times 16d + 32 \times 16Td$ bits, and can be large for large $T$. As will be shown in Section 4, empirically using shared statistics performs similarly as using separate statistics on language modeling tasks, but worse on (permuted) sequential MNIST classification.*

## 4  Experiments

Experiments are performed on character/word-level language modeling and sequential MNIST classification. We compare with the full-precision LSTM, and popular state-of-the-art quantized LSTMs including (i) 1-bit LSTMs with/without normalization: binarized using BinaryConnect (BC) [5], binary weight network (BWN) [23], and loss-aware binarization (LAB) [11]. We also compare

Table 2: Test BPC and size (in KB) of LSTM on character-level language modeling. "N/A" means that the loss become NaN after the first epoch. Method with the lowest BPC in each group is highlighted.

| precision | quantization | normalization | War and Peace BPC | size | Penn Treebank BPC | size | Text8 BPC | size |
|---|---|---|---|---|---|---|---|---|
| full | - | - | 1.72 | 4800 | 1.45 | 4504 | 1.46 | 63375 |
| | | weight | 1.73 | 4816 | 1.45 | 4520 | 1.48 | 63438 |
| | | layer | **1.69** | 4832 | **1.43** | 4536 | **1.45** | 63500 |
| | | batch (shared) | 1.72 | 4864 | 1.45 | 4568 | 1.46 | 63625 |
| | | batch (separate) | 1.72 | 8032 | 1.45 | 7736 | 1.46 | 86000 |
| 1-bit | SBN | batch (separate) | 1.78 | 3794 | 1.60 | 3785 | 1.54 | 27464 |
| | BinaryConnect | - | 4.24 | 158 | 2.51 | 149 | N/A | 2011 |
| | | weight | 1.74 | 174 | 1.50 | 165 | 1.50 | 2073 |
| | | layer | **1.69** | 190 | **1.49** | 181 | **1.47** | 2136 |
| | | batch (shared) | 1.72 | 222 | 1.51 | 213 | **1.47** | 2261 |
| | | batch (separate) | 1.72 | 3390 | 1.50 | 3381 | 1.48 | 24636 |
| | BWN | - | 1.89 | 158 | 1.56 | 149 | 1.56 | 2011 |
| | | weight | 1.74 | 174 | 1.51 | 165 | 1.50 | 2073 |
| | | layer | **1.70** | 190 | **1.49** | 181 | **1.47** | 2136 |
| | | batch (shared) | 1.72 | 222 | 1.50 | 213 | **1.47** | 2261 |
| | | batch (separate) | 1.72 | 3390 | 1.51 | 3381 | 1.48 | 24636 |
| | LAB | - | 1.86 | 158 | 1.56 | 149 | 1.58 | 2011 |
| | | weight | 1.73 | 174 | 1.51 | 165 | 1.50 | 2073 |
| | | layer | **1.70** | 190 | **1.49** | 181 | **1.47** | 2136 |
| | | batch (shared) | 1.71 | 222 | 1.50 | 213 | **1.47** | 2261 |
| | | batch (separate) | 1.72 | 3390 | 1.50 | 3381 | **1.47** | 24636 |
| 2-bit | STN | batch (separate) | 1.72 | 3944 | 1.60 | 3521 | 1.51 | 15303 |
| | TerConnect | - | 6.35 | 308 | 5.84 | 289 | N/A | 3990 |
| | | weight | 1.72 | 324 | **1.42** | 305 | **1.42** | 4053 |
| | | layer | **1.67** | 340 | 1.43 | 321 | 1.44 | 4115 |
| | | batch (shared) | 1.70 | 372 | 1.44 | 353 | 1.44 | 4240 |
| | | batch (separate) | 1.71 | 3540 | 1.45 | 3521 | 1.44 | 26615 |
| | TWN | - | 1.86 | 308 | 1.51 | 289 | 1.54 | 2990 |
| | | weight | 1.71 | 324 | 1.45 | 305 | **1.43** | 4053 |
| | | layer | **1.67** | 340 | **1.43** | 321 | 1.44 | 4115 |
| | | batch (shared) | 1.70 | 372 | 1.44 | 353 | 1.44 | 4240 |
| | | batch (separate) | 1.70 | 3540 | 1.45 | 3521 | 1.44 | 26615 |
| | LAT | - | 1.80 | 308 | 1.48 | 289 | 1.50 | 2990 |
| | | weight | 1.69 | 324 | 1.42 | 305 | 1.44 | 4053 |
| | | layer | **1.65** | 340 | **1.40** | 321 | **1.41** | 4115 |
| | | batch (shared) | 1.68 | 372 | 1.41 | 353 | **1.41** | 4240 |
| | | batch (separate) | 1.68 | 3540 | 1.42 | 3521 | **1.41** | 26615 |

with the recent stochastically binarized LSTM (denoted SBN) with batch normalization in [1]; (ii) 2-bit LSTMs with/without normalization: ternarized using ternary weight networks (TWN) [17], and loss-aware ternarization with approximate solution (LAT) [12]. Analogous to BinaryConnect, we also include a baseline called TerConnect[1], which ternarizes weights to $\{-1, 0, +1\}$ using the same threshold as TWN, but does not scale the ternary weights. We also compare with 2-bit alternating LSTM [29], and the stochastically ternarized LSTM (denoted STN) with batch normalization [1].

## 4.1 Character-level Language Modeling

The LSTM takes as input a character sequence, and predicts the next character at each time step. The training objective is the cross-entropy loss over target sequences, and performance is evaluated by bits per character (BPC). Experiments are performed on three benchmark data sets: (i) Leo Tolstoy's *War and Peace*; (ii) *Penn Treebank* Corpus [27]; and (iii) *Text8*. On *War and Peace* and *Penn Treebank*, we use a one-layer LSTM with 512 hidden units[2] as in [11, 12]. On *text8*, we use a one-layer LSTM with 2000 hidden units as in [1]. Adam is used as the optimizer. The detailed setup is in Appendix A.1.

Table 2 shows the testing BPC values and size of LSTM parameters, including the additional storage due to normalization parameters and statistics (where applicable). Note that [1] does not count this additional storage.

**Normalization:** For the quantized LSTM, the normalized version consistently outperforms its unnormalized counterpart. In particular, directly applying BinaryConnect achieves very poor performance on *War and Peace* and *Penn TreeBank*, and fails on *Text8*. With weight / layer / batch (shared) normalization, BinaryConnect achieves comparable or even better results than the full-precision

LSTM, while being over 20x smaller. For batch normalization, using shared statistics across all time steps yields similar performance as when separate statistics are used, but with much smaller storage.

**Comparison with Full-Precision LSTM:** The 1-bit normalized LSTM performs similarly as the unnormalized full-precision baseline on *War and Peace* and *Text8*. The 2-bit normalized LSTM significantly outperforms the unnormalized full-precision baseline on all three data sets, and requires much less storage. Moreover, compared with the normalized full-precision LSTM, 2-bit normalized LSTM has competitive performance, but requires much smaller storage.

**Comparison of Different Quantization Methods:** For the unnormalized 1-bit LSTM, BWN and LAB perform significantly better than BinaryConnect. They have an additional scaling parameter which is empirically smaller than 1 (as can be seen in Appendix B), and can thus alleviate the exploding gradient problem (Section 2.2). As BWN and LAB differ from BinaryConnect only in the scaling parameter, the three perform similarly when normalization is applied, as the normalized LSTM is invariant to weight rescaling. For the unnormalized 2-bit quantized LSTMs, TWN and LAT also perform significantly better than TerConnect.

**Comparison with SBN and STN:** Directly applying normalization on top of quantization consistently outperforms the batch-normalized LSTM in [1]. In particular, 1-bit layer-normalized BinaryConnect achieves similar and often better performance than 2-bit STN. Moreover, binarized/ternarized model with weight/layer/batch (shared) normalization is significantly smaller than SBN and STN, which use separate statistics for different time steps and normalization on the cell.

$\lambda_1, \lambda_2$ **Values:** Figures 1(a)-1(b) show $\lambda_1, \lambda_2$ (in Propositions 2.1, 3.1-3.2) for the unnormalized full-precision and BC-binarized LSTMs with weight/layer normalization on *Penn Treebank*, and Figures 1(c)-1(d) show the values (in Propositions 3.3-3.4) with batch normalization.[3] As can be seen, normalization reduces $\lambda_1, \lambda_2$ in the binarized LSTM. The corresponding $g_*$ values are in Appendix C.

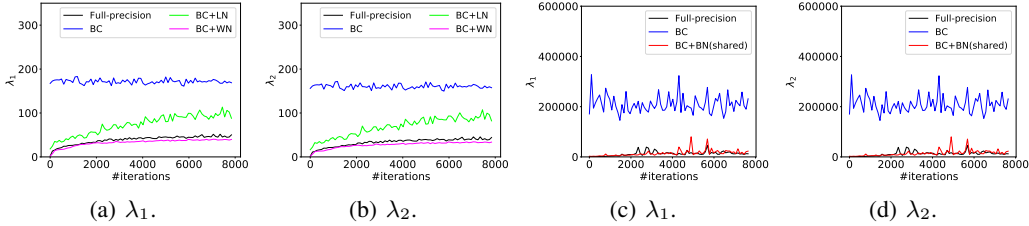

(a) $\lambda_1$.　　　　(b) $\lambda_2$.　　　　(c) $\lambda_1$.　　　　(d) $\lambda_2$.

Figure 1: $\lambda_1$ and $\lambda_2$ values in full-precision and binarized LSTMs (with and without normalization).

**Gradient Magnitude:** Figure 2 shows the backpropagated gradient norms[4] for the unnormalized full-precision LSTM, and BC-normalized LSTM with/without normalization. The gradients of the unnormalized binarized LSTM explode quickly during backpropagation (Figure 2(b)), while the normalized binarized LSTM has stable gradient flow similar to the full-precision baseline. More results can be found in Appendix D.1.

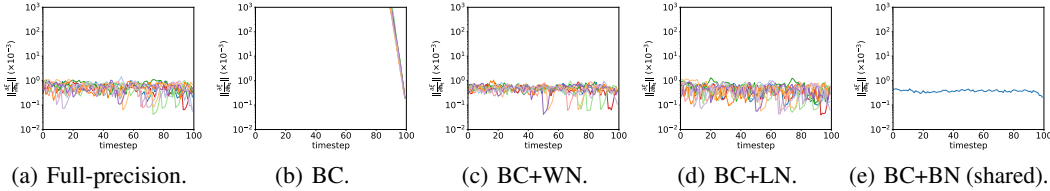

(a) Full-precision.　　(b) BC.　　(c) BC+WN.　　(d) BC+LN.　　(e) BC+BN (shared).

Figure 2: Gradient norms of the unnormalized full-precision LSTM and BC-binarized LSTM with/without normalization for character-level language modeling on *Penn Treebank*. Since backpropagation operates in the backward direction, each plot is best read from right to left.

## 4.2 Word-level Language Modeling

In this section, we perform experiments to predict the next word on *Penn Treebank*. We use a one-layer LSTM, with $d = 300$ as in [1, 29], and $d = 650$ as in [1, 30]. We use the same data preparation and training procedures as in [1, 20]. The optimizer is SGD. Detailed setup is in Appendix A.2.

Table 3 shows the testing perplexity (PPL) results. BinaryConnect and TerConnect fail when directly applied. However, with normalization, they achieve comparable performance as the full-precision counterpart. Quantized models with normalization usually outperform their unnormalized counterparts, and have comparable or even better performance than the full-precision baseline. Again, directly applying normalization to existing quantization methods on LSTM perform similarly or even better than SBN and STN, while requiring much smaller storage when normalized using weight/layer/batch (shared) normalization. For batch normalization, using shared mean and variance statistics across all time steps performs similarly as using separate statistics. Preliminary results on a 2-layer LSTM also show similar observations (details are in Appendix E).

Table 3: Test PPL and size (in KB) of LSTM for word-level language modeling on *Penn Treebank*. For the alternating LSTM, only $d = 300$ are reported in [29].

| precision | quantlization | normalization | $d = 300$ PPL | $d = 300$ size | $d = 650$ PPL | $d = 650$ size |
|---|---|---|---|---|---|---|
| full | - | - | 91.5 | 2817 | 87.6 | 13213 |
| | | weight | **86.1** | 2827 | 86.2 | 13234 |
| | | layer | 87.4 | 2836 | **84.5** | 13254 |
| | | batch (shared) | 90.2 | 2855 | 86.3 | 13295 |
| | | batch (separate) | 90.5 | 3492 | 87.9 | 14676 |
| 1-bit | SBN | batch (separate) | 92.2 | 852 | 87.2 | 2068 |
| | BinaryConnect | - | 8247.4 | 93 | 1244.2 | 423 |
| | | weight | **87.6** | 102 | 84.8 | 443 |
| | | layer | 89.4 | 111 | **82.3** | 463 |
| | | batch (shared) | 92.4 | 130 | 84.8 | 504 |
| | | batch(separate) | 91.9 | 767 | 85.6 | 1885 |
| | BWN | - | 94.7 | 93 | 83.5 | 423 |
| | | weight | **89.4** | 102 | 85.9 | 443 |
| | | layer | 91.4 | 111 | **84.2** | 463 |
| | | batch (shared) | 91.5 | 130 | 86.6 | 504 |
| | | batch (separate) | 93.0 | 767 | 87.3 | 1885 |
| 2-bit | alternating LSTM | - | 103.1 | 180 | - | - |
| | STN | batch (separate) | 90.7 | 940 | 86.1 | 2481 |
| | TerConnect | - | 113.8 | 180 | 113.8 | 835 |
| | | weight | **86.5** | 190 | 84.9 | 856 |
| | | layer | 88.2 | 199 | **82.5** | 876 |
| | | batch (shared) | 90.6 | 218 | 85.8 | 917 |
| | | batch (separate) | 91.6 | 855 | 86.5 | 2298 |
| | TWN | - | 89.8 | 180 | 84.2 | 835 |
| | | weight | **87.1** | 190 | 85.6 | 856 |
| | | layer | 90.5 | 199 | **84.1** | 876 |
| | | batch (shared) | 92.1 | 218 | 85.5 | 917 |
| | | batch (separate) | 91.2 | 855 | 87.5 | 2298 |
| 3-bit | alternating LSTM | - | 93.8 | 268 | - | - |
| 4-bit | alternating LSTM | - | 91.4 | 356 | - | - |

Figure 3 shows the norms of backpropagated gradients in the full-precision and binarized LSTMs. As can be seen, the gradients of the unnormalized binarized LSTM explode quickly during backpropagation (Figure 3(b)), while the normalized binarized LSTM has stable gradient flow similar to the full-precision baseline. This agrees with Propositions 2.2-2.3 and Table 1 that the spectral norm of weight matrix becomes larger (i) for large $d$, and (ii) for BinaryConnect than TerConnect, leading to more severe exploding gradient problem. More results can be found in Appendix D.2.

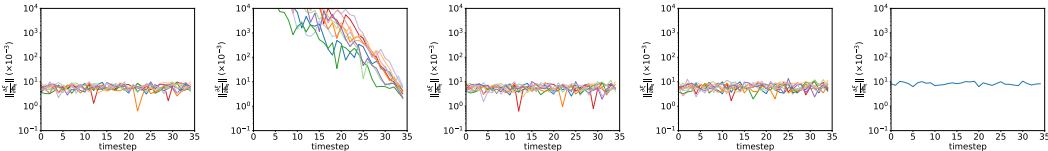

(a) Full-precision.  (b) BC.  (c) BC+WN.  (d) BC+LN.  (e) BC+BN (shared).

Figure 3: Gradient norms of the unnormalized full-precision LSTM and BC-binarized LSTM with/without normalization for word-level language modeling (with $d = 300$) on *Penn Treebank*.

### 4.3 Sequential MNIST

We perform experiments on the sequential version of MNIST classification, which processes one image pixel at a time. We follow the setting in [16, 4, 1], and use both the MNIST and permuted MNIST (pMNIST) [4]. In MNIST, the $28 \times 28$ pixels are processed in scanline order. In pMNIST, they are processed in a fixed random order. The optimizer is Adam. Detailed setup is in Appendix A.3.

Table 4 shows the test accuracy results and size[5] of the LSTM parameters. BinaryConnect and TerConnect, which cannot be trained without normalization on this task, have comparable results as the full-precision baselines when used with normalization.

Batch normalization with shared mean and variance statistics across all time steps has inferior performance; while storing separate mean and variance statistics for the $28 \times 28 = 784$ time steps is too memory expensive. In contrast, weight and layer normalization achieve high model compression, and with comparable or even better performance as the batch normalization counterpart.

Though batch normalization with shared statistics performs similarly as using separate statistics on language modeling tasks (Tables 2-3), it fails on this sequential MNIST task. We speculate it is because in this task, each time step corresponds to an input pixel. The use of shared batch normalization statistics implicitly assumes different pixels to have similar characteristics. However, this may not be reasonable (e.g., pixels around the edge are typically darker).

Table 4: Test accuracy (%) and size (KB) of LSTM on the sequential MNIST task. "N/A" means that the loss becomes NaN after the first epoch.

| precision | quantization | normalization | MNIST | pMNIST | size |
|---|---|---|---|---|---|
| full | - | - | 98.9 | 90.2 | 159 |
| | | weight | 98.4 | 90.2 | 163 |
| | | layer | 98.0 | 90.7 | 166 |
| | | batch (shared) | 21.4 | 35.4 | 172 |
| | | batch (separate) | **99.0** | **93.7** | 5066 |
| 1-bit | SBN | batch (separate) | 98.6 | 89.9 | 5526 |
| | BinaryConnect | - | N/A | N/A | 8 |
| | | weight | 98.7 | **91.4** | 11 |
| | | layer | **98.9** | 91.2 | 14 |
| | | batch (shared) | 20.6 | 36.5 | 21 |
| | | batch (separate) | 98.7 | 91.2 | 4914 |
| | BWN | - | 98.7 | 89.7 | 8 |
| | | weight | 98.7 | **91.3** | 11 |
| | | layer | **98.8** | 90.8 | 14 |
| | | batch (shared) | 20.6 | 40.1 | 21 |
| | | batch (separate) | 98.6 | 91.1 | 4914 |
| 2-bit | STN | batch (separate) | 98.8 | 91.9 | 5531 |
| | TerConnect | - | N/A | N/A | 13 |
| | | weight | **98.9** | 92.4 | 16 |
| | | layer | 98.8 | 92.5 | 19 |
| | | batch (shared) | 23.6 | 34.1 | 25 |
| | | batch (separate) | 98.8 | **93.2** | 4919 |
| | TWN | - | 98.6 | 90.4 | 13 |
| | | weight | 98.6 | 92.1 | 16 |
| | | layer | **98.8** | 91.7 | 18 |
| | | batch (shared)) | 26.5 | 38.3 | 25 |
| | | batch (separate) | 98.7 | **93.1** | 4919 |

## 5 Conclusion

In this paper, we show that quantized LSTMs are hard to train because the scales of the quantized LSTM weights can be very large, making the gradients easy to explode. We then show that applying weight, layer or batch normalization can enable the gradient magnitude to be invariant to this possibly large scaling, and thus alleviates the exploding gradient problem. Experiments on various tasks show that the normalized quantized LSTM can be easily trained, achieves comparable or even better performance than its full-precision counterpart, but saves much storage due to quantization.

**Acknowledgments**

This research project is partially funded by Microsoft Research Asia.

## Footnotes

[1]Note that this is different from the stochastic ternary-connect in [19]

[2]Ardakani *et al.* [1] use 1000 hidden units, so their BPC results are not directly comparable.

[3]Propositions 3.3 and 3.4 are based on the squared weight norm. Hence, Figures 1(c)-1(d) plot the coefficients before $\sum_{k=1}^{N} \|\frac{\partial \xi_m}{\partial \mathbf{h}_t^k}\|^2$ and $\sum_{k=1}^{N} \|\frac{\partial \xi_m}{\partial \mathbf{c}_t^{k+1}}\|^2$. With an abuse of notation, we still call these $\lambda_1$ and $\lambda_2$.

[4]Figures 2(a)-2(d) show $\|\frac{\partial \xi}{\partial \mathbf{h}_t}\|$. Each curve corresponds to the gradient from one sample in the mini-batch. For better visualization, we only show 10 curves. Figure 2(e) shows $\sum_{k=1}^{N} \|\frac{\partial \xi}{\partial \mathbf{h}_t^k}\|^2$ for the whole mini-batch.

[5]Note that [1] does not count the additional storage for batch statistics, which is indeed much larger than the model itself on this sequential MNIST task (where the number of time steps is $T = 784$).

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
