[Supplementary Material]

# A    Experiment Setup

We run all the experiments with PyTorch. The following describes the detailed setup. All the data sets and source code can be found at `https://github.com/houlu369/Normalized-Quantized-LSTM`.

## A.1    Character-level Language Modeling

The Leo Tolstoy's *War and Peace* data set consists of 3258K characters of almost entirely English text, with minimal markup and a vocabulary size of 87. We follow the setup in [11, 12], and use the same training/validation/test set split. We use a one-layer LSTM with 512 hidden units. The maximum number of epochs is 200, and the number of time steps is 100. Batch size is 64. The initial learning rate is 0.002. After 10 epochs, the learning rate is decayed by a factor of 0.98 after each epoch as in [11].

The *Penn Treebank* [27] data set has been frequently used for language modeling. It contains 50 different characters, including English characters, numbers, and punctuations. We follow the setup in [11, 12], and use 5,017K characters for training, 393K for validation, and 442K characters for testing. We use a one-layer LSTM with 512 hidden units. The maximum number of epochs is 200, and the number of time steps is 100. Batch size is 64. The initial learning rate is 0.002. After 10 epochs, the learning rate is decayed by a factor of 0.98 after each epoch as in [11]. Since only results with 1000 hidden units are reported for SBN and STN in the original paper [1], the results for SBN and STN with 512 hidden units in Table 2 are rerun using the released code in [1].

The *Text8* data set has a vocabulary size of 27 and consists of 100M characters. We follow the setup in [1], and split the data into training, validation and test sets with 90M, 5M and 5M characters, respectively. A one-layer LSTM with 2000 hidden units is used. The maximum number of epochs is 100, and the number of time steps is 180. A constant learning rate 0.001 is used. Batch size is 128.

For all three data sets, the gradient norm is clipped at 5 after each iteration. The weights are initialized using the default uniform initialization for LSTM in PyTorch. Adam [15] is used as the optimizer. We initialize the hidden states to zero, and use the final hidden states of the current minibatch as the initial hidden state of the subsequent minibatch.

## A.2    Word-level Language Modeling

We split the *Penn Treebank* corpus with a 10K size vocabulary, resulting in 929K training, 73K validation, and 82K test tokens. The batch size is 20. The learning rate is initialized to 20, and divided by 4 when there is an increase in the validation perplexity value. The model is trained with a word sequence length of 35 and dropout probability of 0.5. SGD is used to train the model. The gradient norm is clipped at 0.25. The weights are initialized using the default uniform initialization in PyTorch. We initialize the hidden states to zero, and use the final hidden states of the current minibatch as the initial hidden state of the subsequent minibatch. For BinaryConnect and TerConnect, we scale the learning rate for the eight weight matrices in LSTM by $2/a$, where the weights are initialized uniformly in $[-a, a]$. A similar learning rate scaling is used in BinaryConnect [5].

## A.3    Sequential MNIST

MNIST contains $28 \times 28$ gray-scale images from ten digit classes. We use $50,000$ images for training, another $10,000$ for validation, and the remaining $10,000$ for testing. A one-layer LSTM with 100 hidden units is used, along with a softmax classifier layer. We use the Adam optimizer. The maximum number of epochs is 200. The learning rate decays exponentially from 0.002 to 0.001. Following [4], the gradient norm is clipped at 1. The weights are initialized as in [4]. Since the input at each time step is only one scalar and $\mathbf{W}_{xi}, \mathbf{W}_{xf}, \mathbf{W}_{xa}, \mathbf{W}_{xo} \in \mathbb{R}^{d \times 1}$, we do not quantize these weights as doing so does not have clear advantage to storage and computation. Results for the full-precision baseline, SBN and STN are taken from [1].

# B   Scaling Parameters

As discussed in Section 2.2, we speculate that the empirical success of quantization methods with scaling parameters (like BWN and LAB) is that they use a small scaling parameter $\alpha$ (e.g., smaller than 1) which reduces the norm of $\mathbf{W}_{h*}$. Figure 4 shows the scaling parameter $\alpha$ of $\mathbf{W}_{h*}$ of BWN and LAB on both character-level and word-level language modeling tasks on *Penn Treebank*. As can be seen, the scaling parameters learned are smaller than 1.

(a) BWN (char-level).    (b) LAB (char-level).    (c) BWN (word-level).    (d) LAB (word-level).

Figure 4: Scaling parameter $\alpha$ for BWN and LAB for character- and word-level language modeling on *Penn Treebank*.

# C   Visualizations of $g_*$

Figure 5 shows the $g_*$ values for the binarized LSTM using BinaryConnect with different normalizations, for character-level language modeling on *Penn Treebank*. As can be seen, the $g_*$ value only increases slightly during training.

(a) WN.                    (b) LN.                    (c) BN.

Figure 5: $g$ values for the binarized LSTM using BinaryConnect with different normalizations, for character-level language modeling on *Penn Treebank*.

# D   Norm of Backpropagated Gradient

## D.1   Character-level Language Modeling

Figures 6-8 show more results on the norm of backpropagated gradient. Similar to the observations in Section 4.1, gradients of the original unnormalized LSTM explode quickly, while the normalized counterpart has stable gradient flow.

(a) Full-precision.    (b) BC.    (c) BC+WN.    (d) BC+LN.    (e) BC+BN (shared).

Figure 6: Gradient norms of the unnormalized full-precision LSTM and BC-binarized LSTM with/without normalization for character-level language modeling on *War and Peace*.

| (a) Full-precision. | (b) TC. | (c) TC+WN. | (d) TC+LN. | (e) TC+BN (shared). |

Figure 7: Gradient norms of the unnormalized full-precision LSTM and TC-ternarized LSTM with/without normalization for character-level language modeling on *War and Peace*.

| (a) Full-precision. | (b) TC. | (c) TC+WN. | (d) TC+LN. | (e) TC+BN (shared). |

Figure 8: Gradient norms of the unnormalized full-precision LSTM and TC-ternarized LSTM with/without normalization for character-level language modeling on *Penn Treebank*.

## D.2 Word-level Language Modeling

Figures 9-11 show more results on the norm of the backpropagated gradient. As can be seen, the gradient of binarized LSTM explodes quickly. For ternarized LSTM, gradient explodes at a slower speed for large $d$, and does not explode for small $d$.

| (a) Full-precision. | (b) TC. | (c) TC+WN. | (d) TC+LN. | (e) TC+BN (shared). |

Figure 9: Gradient norms of the unnormalized full-precision LSTM and TC-ternarized LSTM with/without normalization for word-level language modeling (with $d = 300$) on *Penn Treebank*.

| (a) Full-precision. | (b) BC. | (c) BC+WN. | (d) BC+LN. | (e) BC+BN (shared). |

Figure 10: Gradient norms of the unnormalized full-precision LSTM and BC-binarized LSTM with/without normalization for word-level language modeling (with $d = 650$) on *Penn Treebank*.

| (a) Full-precision. | (b) TC. | (c) TC+WN. | (d) TC+LN. | (e) TC+BN (shared). |

Figure 11: Gradient norms of the unnormalized full-precision LSTM and TC-ternarized LSTM with/without normalization for word-level language modeling (with $d = 650$) on *Penn Treebank*.

# E  Multi-layer LSTM

Table 5 shows the results on a 2-layer LSTM, with 650 hidden units per layer, for word-level language modeling on *Penn Treebank*. Again, BinaryConnect/TerConnect fail when directly used, but achieve results comparable as the full-precision baseline when normalized, while being much smaller in size.

Table 5: Test PPL and size (KB) of 2-layer LSTM on word-level language modeling on *Penn Treebank*.

| precision | quantlization | normalization | PPL | size |
|---|---|---|---|---|
| full | - | - | 81.67 | 26427 |
| | | weight | 81.82 | 26467 |
| | | layer | 81.02 | 26508 |
| | | batch (shared) | 80.60 | 26589 |
| | | batch (separate) | 81.42 | 29352 |
| 1-bit | BinaryConnect | - | 134.47 | 846 |
| | | weight | 81.86 | 886 |
| | | layer | 82.10 | 927 |
| | | batch (shared) | 80.55 | 1008 |
| | | batch(separate) | 81.48 | 3771 |
| | BWN | - | 83.12 | 846 |
| | | weight | 83.32 | 886 |
| | | layer | 82.12 | 927 |
| | | batch (shared) | 82.73 | 1008 |
| | | batch (separate) | 82.33 | 3771 |
| 2-bit | TerConnect | - | 639.61 | 1671 |
| | | weight | 80.54 | 1711 |
| | | layer | 80.06 | 1752 |
| | | batch (shared) | 79.25 | 1833 |
| | | batch (separate) | 79.89 | 4596 |
| | TWN | - | 82.13 | 1671 |
| | | weight | 82.14 | 1711 |
| | | layer | 80.98 | 1752 |
| | | batch (shared) | 81.32 | 1833 |
| | | batch (separate) | 81.69 | 4596 |

# F  Proofs

## F.1  Proof for Proposition 2.1

According to [7], we have

$$\frac{\partial \xi_m}{\partial \mathbf{c}_t} = \frac{\partial \xi_m}{\partial \mathbf{h}_t}\text{Diag}(\sigma(\mathbf{o}_t) \odot \tanh'(\mathbf{c}_t)) + \frac{\partial \xi_m}{\partial \mathbf{c}_{t+1}}\text{Diag}(\sigma(\mathbf{f}_{t+1})).$$

$$
\begin{aligned}
\frac{\partial \xi_m}{\partial \mathbf{h}_{t-1}} &= \frac{\partial \xi_m}{\partial \mathbf{c}_t}\frac{\partial \mathbf{c}_t}{\partial \mathbf{h}_{t-1}} + \frac{\partial \xi_m}{\partial \mathbf{o}_t}\frac{\partial \mathbf{o}_t}{\partial \mathbf{h}_{t-1}} \\
&= \frac{\partial \xi_m}{\partial \mathbf{c}_t}\frac{\partial(\sigma(\mathbf{i}_t) \odot \tanh(\mathbf{a}_t) + \sigma(\mathbf{f}_t) \odot \mathbf{c}_{t-1})}{\partial \mathbf{h}_{t-1}} + \frac{\partial \xi_m}{\partial \mathbf{h}_t}\frac{\partial \mathbf{h}_t}{\partial \mathbf{o}_t}\frac{\partial \mathbf{o}_t}{\partial \mathbf{h}_{t-1}} \\
&= \frac{\partial \xi_m}{\partial \mathbf{c}_t}\left(\text{Diag}(\sigma(\mathbf{i}_t))\frac{\partial \tanh(\mathbf{a}_t)}{\partial \mathbf{h}_{t-1}} + \text{Diag}(\tanh(\mathbf{a}_t))\frac{\partial \sigma(\mathbf{i}_t)}{\partial \mathbf{h}_{t-1}} + \text{Diag}(\mathbf{c}_{t-1})\frac{\partial \sigma(\mathbf{f}_t)}{\partial \mathbf{h}_{t-1}}\right) \\
&\quad + \frac{\partial \xi_m}{\partial \mathbf{h}_t}\frac{\partial \mathbf{h}_t}{\partial \mathbf{o}_t}\frac{\partial \mathbf{o}_t}{\partial \mathbf{h}_{t-1}} \\
&= \frac{\partial \xi_m}{\partial \mathbf{c}_t}\left(\text{Diag}(\tanh(\mathbf{a}_t) \odot \sigma'(\mathbf{i}_t))\frac{\partial \mathbf{i}_t}{\partial \mathbf{h}_{t-1}} + \text{Diag}(\mathbf{c}_{t-1} \odot \sigma'(\mathbf{f}_t))\frac{\partial \mathbf{f}_t}{\partial \mathbf{h}_{t-1}}\right. \\
&\quad \left. + \text{Diag}(\sigma(\mathbf{i}_t) \odot \tanh'(\mathbf{a}_t))\frac{\partial \mathbf{a}_t}{\partial \mathbf{h}_{t-1}}\right) + \frac{\partial \xi_m}{\partial \mathbf{h}_t}\text{Diag}(\tanh(\mathbf{c}_t) \odot \sigma'(\mathbf{o}_t))\frac{\partial \mathbf{o}_t}{\partial \mathbf{h}_{t-1}}.
\end{aligned}
$$

Combining the above two, we have

$$
\begin{aligned}
\frac{\partial \xi_m}{\partial \mathbf{h}_{t-1}} &= \frac{\partial \xi_m}{\partial \mathbf{c}_t}\left(\mathrm{Diag}(\tanh(\mathbf{a}_t) \odot \sigma'(\mathbf{i}_t))\mathbf{W}_{hi} + \mathrm{Diag}(\mathbf{c}_{t-1} \odot \sigma(\mathbf{f}_t))\mathbf{W}_{hf}\right. \\
&\quad \left. + \mathrm{Diag}(\sigma(\mathbf{i}_t) \odot \tanh'(\mathbf{a}_t))\mathbf{W}_{ha}\right) + \frac{\partial \xi_m}{\partial \mathbf{h}_t}\mathrm{Diag}(\tanh(\mathbf{c}_t) \odot \sigma'(\mathbf{o}_t))\mathbf{W}_{ho} \\[2mm]
&= \frac{\partial \xi_m}{\partial \mathbf{h}_t}\left(\underbrace{\mathrm{Diag}(\sigma(\mathbf{o}_t) \odot \tanh'(\mathbf{c}_t) \odot \tanh(\mathbf{a}_t) \odot \sigma'(\mathbf{i}_t))}_{\mathbf{s}_t^i} \mathbf{W}_{hi}\right. \\
&\quad + \mathrm{Diag}(\mathbf{c}_{t-1})\underbrace{\mathrm{Diag}(\sigma(\mathbf{o}_t) \odot \tanh'(\mathbf{c}_t) \odot \sigma'(\mathbf{f}_t))}_{\mathbf{s}_t^f} \mathbf{W}_{hf} \\
&\quad + \underbrace{\mathrm{Diag}(\sigma(\mathbf{o}_t) \odot \tanh'(\mathbf{c}_t) \odot \sigma(\mathbf{i}_t) \odot \tanh'(\mathbf{a}_t))}_{\mathbf{s}_t^a} \mathbf{W}_{ha} \\
&\quad \left. + \underbrace{\mathrm{Diag}(\tanh(\mathbf{c}_t) \odot \sigma'(\mathbf{o}_t))}_{\mathbf{s}_t^o} \mathbf{W}_{ho}\right) \\
&\quad + \frac{\partial \xi_m}{\partial \mathbf{c}_{t+1}}\left(\underbrace{\mathrm{Diag}(\sigma(\mathbf{f}_{t+1}) \odot \tanh(\mathbf{a}_t) \odot \sigma'(\mathbf{i}_t))}_{\mathbf{u}_t^i} \mathbf{W}_{hi}\right. \\
&\quad + \mathrm{Diag}(\mathbf{c}_{t-1})\underbrace{\mathrm{Diag}(\sigma(\mathbf{f}_{t+1}) \odot \sigma'(\mathbf{f}_t))}_{\mathbf{u}_t^f} \mathbf{W}_{hf} \\
&\quad \left. + \underbrace{\mathrm{Diag}(\sigma(\mathbf{f}_{t+1}) \odot \sigma(\mathbf{i}_t) \odot \tanh'(\mathbf{a}_t))}_{\mathbf{u}_t^a} \mathbf{W}_{ha}\right).
\end{aligned}
$$

Since the sigmoid, tanh functions and their derivatives are bounded,

$$
\sigma(x) \in (0,1), \tanh(x) \in (-1,1), \sigma'(x) = \sigma(x)(1-\sigma(x)) \in (0,\tfrac{1}{4}], \tanh'(x) = \in (0,1]. \quad (8)
$$

It is easy to verify that

$$
\begin{aligned}
\|\mathbf{s}_t^i\|_2 &= \|\mathrm{Diag}(\sigma(\mathbf{o}_t) \odot \tanh'(\mathbf{c}_t) \odot \tanh(\mathbf{a}_t) \odot \sigma'(\mathbf{i}_t))\|_2 \\
&= \|\sigma(\mathbf{o}_t) \odot \tanh'(\mathbf{c}_t) \odot \tanh(\mathbf{a}_t) \odot \sigma'(\mathbf{i}_t)\|_\infty \\
&< 1 \times 1 \times 1 \times \frac{1}{4} \\
&< \frac{1}{4}.
\end{aligned}
$$

Similarly,

$$
\|\mathbf{s}_t^f\|_2 < \frac{1}{4}, \|\mathbf{s}_t^a\|_2 < 1, \|\mathbf{s}_t^o\|_2 < \frac{1}{4}, \|\mathbf{u}_t^i\|_2 < \frac{1}{4}, \|\mathbf{u}_t^f\|_2 < \frac{1}{4}, \|\mathbf{u}_t^a\|_2 < 1.
$$

Using the property of norms, we have

$$
\begin{aligned}
\left\|\frac{\partial \xi_m}{\partial \mathbf{h}_{t-1}}\right\|_2 &\leq \left(\frac{1}{4}\|\mathbf{W}_{hi}\|_2 + \frac{1}{4}\gamma_1\|\mathbf{W}_{hf}\|_2 + \|\mathbf{W}_{ha}\|_2 + \frac{1}{4}\|\mathbf{W}_{ho}\|_2\right)\left\|\frac{\partial \xi_m}{\partial \mathbf{h}_t}\right\|_2 \\
&\quad + \left(\frac{1}{4}\|\mathbf{W}_{hi}\|_2 + \frac{1}{4}\gamma_1\|\mathbf{W}_{hf}\|_2 + \|\mathbf{W}_{ha}\|_2\right)\left\|\frac{\partial \xi_m}{\partial \mathbf{c}_{t+1}}\right\|_2.
\end{aligned}
$$

### F.2 Proof for Proposition 3.1

**Proof 1** *For weight normalization $\mathbf{y} = \mathcal{WN}(\mathbf{W}\mathbf{x})$, the Jacobian $\frac{\partial \mathbf{y}}{\partial \mathbf{x}}$ can be written as*

$$\frac{\partial \mathbf{y}}{\partial \mathbf{x}} = Diag(\mathbf{g})\mathbf{D}^{-1}\mathbf{W}, \tag{9}$$

*where $\mathbf{D} = Diag([\|\mathbf{W}_1\|, \|\mathbf{W}_2\|, \cdots, \|\mathbf{W}_d\|]^\top)$, with $\mathbf{W}_i$ the ith row of $\mathbf{W}$.*

*Similar to the derivation in the proof for Proposition 2.1, we have*

$$\frac{\partial \xi_m}{\partial \mathbf{c}_t} = \frac{\partial \xi_m}{\partial \mathbf{h}_t} Diag(\sigma(\tilde{\mathbf{o}}_t) \odot \tanh'(\mathbf{c}_t)) + \frac{\partial \xi_m}{\partial \mathbf{c}_{t+1}} Diag(\sigma(\tilde{\mathbf{f}}_{t+1})). \tag{10}$$

$$
\begin{aligned}
\frac{\partial \xi_m}{\partial \mathbf{h}_{t-1}} =\;& \frac{\partial \xi_m}{\partial \mathbf{c}_t} \left( Diag(tanh(\tilde{\mathbf{a}}_t) \odot \sigma'(\tilde{\mathbf{i}}_t)) \frac{\partial \tilde{\mathbf{i}}_t}{\partial \mathbf{h}_{t-1}} + Diag(\mathbf{c}_{t-1} \odot \sigma'(\tilde{\mathbf{f}}_t)) \frac{\partial \tilde{\mathbf{f}}_t}{\partial \mathbf{h}_{t-1}} \right. \\
&\left. + Diag(\sigma(\tilde{\mathbf{i}}_t) \odot tanh'(\tilde{\mathbf{a}}_t)) \frac{\partial \tilde{\mathbf{a}}_t}{\partial \mathbf{h}_{t-1}} \right) + \frac{\partial \xi_m}{\partial \mathbf{h}_t} Diag(tanh(\mathbf{c}_t) \odot \sigma'(\tilde{\mathbf{o}}_t)) \frac{\partial \tilde{\mathbf{o}}_t}{\partial \mathbf{h}_{t-1}}.
\end{aligned}
$$

*Combining the above two, we have*

$$
\begin{aligned}
\frac{\partial \xi_m}{\partial \mathbf{h}_{t-1}} =\;& \frac{\partial \xi_m}{\partial \mathbf{h}_t} \left( Diag(\sigma(\tilde{\mathbf{o}}_t) \odot tanh'(\mathbf{c}_t) \odot tanh(\tilde{\mathbf{a}}_t) \odot \sigma'(\tilde{\mathbf{i}}_t)) \frac{\partial \tilde{\mathbf{i}}_t}{\partial \mathbf{h}_{t-1}} \right. \\
&+ Diag(\mathbf{c}_{t-1}) Diag(\sigma(\tilde{\mathbf{o}}_t) \odot tanh'(\mathbf{c}_t) \odot \sigma'(\tilde{\mathbf{f}}_t)) \frac{\partial \tilde{\mathbf{f}}_t}{\partial \mathbf{h}_{t-1}} \\
&+ Diag(\sigma(\tilde{\mathbf{o}}_t) \odot tanh'(\mathbf{c}_t)) \odot \sigma(\tilde{\mathbf{i}}_t) \odot tanh'(\tilde{\mathbf{a}}_t)) \frac{\partial \tilde{\mathbf{a}}_t}{\partial \mathbf{h}_{t-1}} \\
&\left. + Diag(tanh(\mathbf{c}_t) \odot \sigma'(\tilde{\mathbf{o}}_t)) \frac{\partial \tilde{\mathbf{o}}_t}{\partial \mathbf{h}_{t-1}} \right) \\
&+ \frac{\partial \xi_m}{\partial \mathbf{c}_{t+1}} \left( Diag(\sigma(\tilde{\mathbf{f}}_{t+1}) \odot tanh(\tilde{\mathbf{a}}_t) \odot \sigma'(\tilde{\mathbf{i}}_t)) \frac{\partial \tilde{\mathbf{i}}_t}{\partial \mathbf{h}_{t-1}} \right. \\
&+ Diag(\mathbf{c}_{t-1}) Diag(\sigma(\tilde{\mathbf{f}}_{t+1}) \odot \sigma'(\tilde{\mathbf{f}}_t)) \frac{\partial \tilde{\mathbf{f}}_t}{\partial \mathbf{h}_{t-1}} \\
&\left. + Diag(\sigma(\tilde{\mathbf{f}}_{t+1}) \odot \sigma(\tilde{\mathbf{i}}_t) \odot tanh'(\tilde{\mathbf{a}}_t)) \frac{\partial \tilde{\mathbf{a}}_t}{\partial \mathbf{h}_{t-1}} \right). \tag{11}
\end{aligned}
$$

*Using the property of norms, (8), (10), and (11) we have*

$$
\begin{aligned}
\left\| \frac{\partial \xi_m}{\partial \mathbf{h}_{t-1}} \right\| \leq\;& \left( \frac{g_i}{4} \|\mathbf{D}_i^{-1}\mathbf{W}_{hi}\|_2 + \frac{\gamma_1 g_f}{4} \|\mathbf{D}_f^{-1}\mathbf{W}_{hf}\|_2 + g_a \|\mathbf{D}_a^{-1}\mathbf{W}_{ha}\|_2 + \frac{g_o}{4} \|\mathbf{D}_o^{-1}\mathbf{W}_{ho}\|_2 \right) \left\| \frac{\partial \xi_m}{\partial \mathbf{h}_t} \right\| \\
&+ \left( \frac{g_i}{4} \|\mathbf{D}_i^{-1}\mathbf{W}_{hi}\|_2 + \frac{\gamma_1 g_f}{4} \left\| \mathbf{D}_f^{-1}\mathbf{W}_{hf} \right\|_2 + g_a \|\mathbf{D}_a^{-1}\mathbf{W}_{ha}\|_2 \right) \left\| \frac{\partial \xi_m}{\partial \mathbf{c}_{t+1}} \right\|.
\end{aligned}
$$

### F.3 Proof for Proposition 3.2

In the following, we first derive the Jacobian of the output after layer normalization w.r.t the input before layer normalization. For layer normalization $\mathbf{y} = \mathcal{LN}(\mathbf{x}) = \mathbf{g} \odot \frac{\mathbf{x} - \mu \mathbf{1}}{\sigma} + \mathbf{b}$ with $\mathbf{z} = \frac{\mathbf{x} - \mu \mathbf{1}}{\sigma}$ the vector with zero mean and unit variance, we have

$$
\begin{aligned}
\frac{\partial y_j}{\partial x_i} =\;& \frac{\partial (g_j \frac{x_j - \mu}{\sigma} + b_j)}{\partial x_i} = g_j \frac{\partial (\frac{x_j - \mu}{\sigma})}{\partial x_i} \\
=\;& g_j \left( \frac{(\mathbf{1}_{i=j} - 1/d)\sigma - (x_j - \mu)1/(2\sigma)(2/d \sum_{i=1}^d (x_i - \mu)(\mathbf{1}_{i=j} - 1/d))}{\sigma^2} \right), \\
=\;& \frac{g_j}{\sigma} \left( \mathbf{1}_{i=j} - \frac{1}{d} - \frac{1}{d} z_i z_j \right).
\end{aligned}
$$

Here, $\mathbf{1}_{i=j}$ is equal to 1 if $i = j$ else 0. Thus the Jacobian is

$$\frac{\partial \mathbf{y}}{\partial \mathbf{x}} = \frac{\mathbf{g}}{\sigma} \odot \left( \mathbf{I} - \frac{1}{d} (\mathbf{1}\mathbf{1}^\top + \mathbf{z}\mathbf{z}^\top) \right), \tag{12}$$

where $\mathbf{1} = [1, 1, \cdots, 1]^\top \in \mathbb{R}^d$. As $\mathbf{z}$ is a vector normalized to zero mean and unit variance, we have $\|\mathbf{z}\mathbf{z}^\top\|_2 = \|\mathbf{z}\mathbf{z}^\top\|_F = d$. Thus

$$\frac{1}{d}\|\mathbf{1}\mathbf{1}^\top + \mathbf{z}\mathbf{z}^\top\|_2 \le \frac{1}{d}(\|\mathbf{1}\mathbf{1}^\top\|_2 + \|\mathbf{z}\mathbf{z}^\top\|_2) \le \frac{1}{d}(d+d) \le 2. \tag{13}$$

Since $\mathbf{1}\mathbf{1}^\top$ and $\mathbf{z}\mathbf{z}^\top$ are both PSDs,

$$\left\| \frac{1}{d}(\mathbf{1}\mathbf{1}^\top + \mathbf{z}\mathbf{z}^\top) \right\|_2 \ge \frac{1}{d}\|\mathbf{1}\mathbf{1}^\top\|_2 = 1. \tag{14}$$

From (13) and (14), the maximum eigenvalue of $\frac{1}{d}(\mathbf{1}\mathbf{1}^\top + \mathbf{z}\mathbf{z}^\top)$ satisfies $1 \le \lambda_{max} \le 2$. Note that all the eigenvalues are non-negative, and $\lambda_{min} \le 0$. For a symmetric matrix $\mathbf{A}$, $\|\mathbf{A}\|_2$ is equal to the square root of maximal eigenvalue of $\mathbf{A}^2$, namely, the maximal absolute eigenvalue of $\mathbf{A}$. Denote $\mathbf{B} = \frac{1}{d}(\mathbf{1}\mathbf{1}^\top + \mathbf{z}\mathbf{z}^\top)$, since $(\mathbf{I} - \frac{1}{d}(\mathbf{1}\mathbf{1}^\top + \mathbf{z}\mathbf{z}^\top))$ is symmetric, we have

$$\begin{aligned}
\left\| (\mathbf{I} - \frac{1}{d}(\mathbf{1}\mathbf{1}^\top + \mathbf{z}\mathbf{z}^\top)) \right\|_2 &= \max_{\|\mathbf{v}\|=1} \left| \frac{\mathbf{v}^\top(\mathbf{I} - \mathbf{B})\mathbf{v}}{\mathbf{v}^\top\mathbf{v}} \right| = \max_{\|\mathbf{v}\|=1} \left\{ 1 - \frac{\mathbf{v}^\top\mathbf{B}\mathbf{v}}{\mathbf{v}^\top\mathbf{v}}, \frac{\mathbf{v}^\top\mathbf{B}\mathbf{v}}{\mathbf{v}^\top\mathbf{v}} - 1 \right\} \\
&= \max\{1 - \lambda_{min}, \lambda_{max} - 1\} \le 1.
\end{aligned}$$

Thus the Jacobian matrix satisfies:

$$\left\| \frac{\partial \mathbf{y}}{\partial \mathbf{x}} \right\|_2 \le \frac{\|\mathrm{Diag}(\mathbf{g})\|_2}{\sigma} = \frac{\max_{1 \le j \le d} g_j}{\sigma}.$$

Then we consider the gradient flow in an LSTM with layer normalization. Similar to the derivation in the proof for Proposition 3.1, we have

$$\frac{\partial \xi_m}{\partial \mathbf{c}_t} = \frac{\partial \xi_m}{\partial \mathbf{h}_t} \mathrm{Diag}(\sigma(\tilde{\mathbf{o}}_t) \odot \tanh'(\mathbf{c}_t)) + \frac{\partial \xi_m}{\partial \mathbf{c}_{t+1}} \mathrm{Diag}(\sigma(\tilde{\mathbf{f}}_{t+1})). \tag{15}$$

$$\begin{aligned}
\frac{\partial \xi_m}{\partial \mathbf{h}_{t-1}} = & \frac{\partial \xi_m}{\partial \mathbf{c}_t} \Big( \mathrm{Diag}(\tanh(\tilde{\mathbf{a}}_t) \odot \sigma'(\tilde{\mathbf{i}}_t)) \frac{\partial \tilde{\mathbf{i}}_t}{\partial \mathbf{h}_{t-1}} + \mathrm{Diag}(\mathbf{c}_{t-1} \odot \sigma'(\tilde{\mathbf{f}}_t)) \frac{\partial \tilde{\mathbf{f}}_t}{\partial \mathbf{h}_{t-1}} \\
& + \mathrm{Diag}(\sigma(\tilde{\mathbf{i}}_t) \odot \tanh'(\tilde{\mathbf{a}}_t)) \frac{\partial \tilde{\mathbf{a}}_t}{\partial \mathbf{h}_{t-1}} \Big) + \frac{\partial \xi_m}{\partial \mathbf{h}_t} \mathrm{Diag}(\tanh(\mathbf{c}_t) \odot \sigma'(\tilde{\mathbf{o}}_t)) \frac{\partial \tilde{\mathbf{o}}_t}{\partial \mathbf{h}_{t-1}}.
\end{aligned}$$

Combining the above two, we have

$$\begin{aligned}
\frac{\partial \xi_m}{\partial \mathbf{h}_{t-1}} = & \frac{\partial \xi_m}{\partial \mathbf{h}_t} \Bigg( \mathrm{Diag}(\sigma(\tilde{\mathbf{o}}_t) \odot \tanh'(\mathbf{c}_t) \odot \tanh(\tilde{\mathbf{a}}_t) \odot \sigma'(\tilde{\mathbf{i}}_t)) \frac{\partial \tilde{\mathbf{i}}_t}{\partial \mathbf{i}_t} \frac{\partial \mathbf{i}_t}{\partial \mathbf{h}_{t-1}} \\
& + \mathrm{Diag}(\mathbf{c}_{t-1}) \mathrm{Diag}(\sigma(\tilde{\mathbf{o}}_t) \odot \tanh'(\mathbf{c}_t) \odot \sigma'(\tilde{\mathbf{f}}_t)) \frac{\partial \tilde{\mathbf{f}}_t}{\partial \mathbf{f}_t} \frac{\partial \mathbf{f}_t}{\partial \mathbf{h}_{t-1}} \\
& + \mathrm{Diag}(\sigma(\tilde{\mathbf{o}}_t) \odot \tanh'(\mathbf{c}_t)) \odot \sigma(\tilde{\mathbf{i}}_t) \odot \tanh'(\tilde{\mathbf{a}}_t)) \frac{\partial \tilde{\mathbf{a}}_t}{\partial \mathbf{a}_t} \frac{\partial \mathbf{a}_t}{\partial \mathbf{h}_{t-1}} \\
& + \mathrm{Diag}(\tanh(\mathbf{c}_t) \odot \sigma'(\tilde{\mathbf{o}}_t)) \frac{\partial \tilde{\mathbf{o}}_t}{\partial \mathbf{h}_{t-1}} \Bigg) \\
& + \frac{\partial \xi_m}{\partial \mathbf{c}_{t+1}} \Bigg( \mathrm{Diag}(\sigma(\tilde{\mathbf{f}}_{t+1}) \odot \tanh(\tilde{\mathbf{a}}_t) \odot \sigma'(\tilde{\mathbf{i}}_t)) \frac{\partial \tilde{\mathbf{i}}_t}{\partial \mathbf{i}_t} \frac{\partial \mathbf{i}_t}{\partial \mathbf{h}_{t-1}} \\
& + \mathrm{Diag}(\mathbf{c}_{t-1}) \mathrm{Diag}(\sigma(\tilde{\mathbf{f}}_{t+1}) \odot \sigma'(\tilde{\mathbf{f}}_t)) \frac{\partial \tilde{\mathbf{f}}_t}{\partial \mathbf{f}_t} \frac{\partial \mathbf{f}_t}{\partial \mathbf{h}_{t-1}} \\
& + \mathrm{Diag}(\sigma(\tilde{\mathbf{f}}_{t+1}) \odot \sigma(\tilde{\mathbf{i}}_t) \odot \tanh'(\tilde{\mathbf{a}}_t)) \frac{\partial \tilde{\mathbf{a}}_t}{\partial \mathbf{a}_t} \frac{\partial \mathbf{a}_t}{\partial \mathbf{h}_{t-1}} \Bigg). \tag{16}
\end{aligned}$$

Using the property of norms, and (8), (15) and (16), we have

$$\left\|\frac{\partial \xi_m}{\partial \mathbf{h}_{t-1}}\right\| \leq \left(\frac{1}{4}\frac{g_i}{\sigma_i}\|\mathbf{W}_{hi}\|_2 + \frac{\gamma_1}{4}\frac{g_f}{\sigma_f}\|\mathbf{W}_{hf}\|_2 + \frac{g_a}{\sigma_a}\|\mathbf{W}_{ha}\|_2 + \frac{1}{4}\frac{g_o}{\sigma_o}\|\mathbf{W}_{ho}\|_2\right)\left\|\frac{\partial \xi_m}{\partial \mathbf{h}_t}\right\|$$

$$+ \left(\frac{1}{4}\frac{g_i}{\sigma_i}\|\mathbf{W}_{hi}\|_2 + \frac{\gamma_1}{4}\frac{g_f}{\sigma_f}\|\mathbf{W}_{hf}\|_2 + \frac{g_a}{\sigma_a}\|\mathbf{W}_{ha}\|_2\right)\left\|\frac{\partial \xi_m}{\partial \mathbf{c}_{t+1}}\right\|.$$

### F.4 Proof for Proposition 3.3

From the proof for Proposition 2.1, we have

$$\frac{\partial \xi_m}{\partial \mathbf{c}_t} = \frac{\partial \xi_m}{\partial \mathbf{h}_t}\text{Diag}(\sigma(\mathbf{o}_t) \odot \tanh'(\mathbf{c}_t)) + \frac{\partial \xi_m}{\partial \mathbf{c}_{t+1}}\text{Diag}(\sigma(\mathbf{f}_{t+1})).$$

Thus

$$\left\|\frac{\partial \xi_m}{\partial \mathbf{c}_t}\right\|^2 \leq \left(\left\|\frac{\partial \xi_m}{\partial \mathbf{h}_t}\right\| + \left\|\frac{\partial \xi_m}{\partial \mathbf{c}_{t+1}}\right\|\right)^2 \leq 2\left\|\frac{\partial \xi_m}{\partial \mathbf{h}_t}\right\|^2 + 2\left\|\frac{\partial \xi_m}{\partial \mathbf{c}_{t+1}}\right\|^2. \qquad (17)$$

Similarly, from the proof for Proposition 2.1, we have

$$\frac{\partial \xi_m}{\partial \mathbf{h}_{t-1}} = \frac{\partial \xi_m}{\partial \mathbf{c}_t}\left(\text{Diag}(\tanh(\mathbf{a}_t) \odot \sigma'(\mathbf{i}_t))\mathbf{W}_{hi} + \text{Diag}(\mathbf{c}_{t-1} \odot \sigma'(\mathbf{f}_t))\mathbf{W}_{hf}\right.$$

$$\left. + \text{Diag}(\sigma(\mathbf{i}_t) \odot \tanh'(\mathbf{a}_t))\mathbf{W}_{ha}\right) + \frac{\partial \xi_m}{\partial \mathbf{h}_t}\text{Diag}(\tanh(\mathbf{c}_t) \odot \sigma'(\mathbf{o}_t))\mathbf{W}_{ho}.$$

$$\left\|\frac{\partial \xi_m}{\partial \mathbf{h}_{t-1}}\right\| \leq \left\|\frac{\partial \xi_m}{\partial \mathbf{c}_t}\right\|\|\text{Diag}(\tanh(\mathbf{a}_t) \odot \sigma'(\mathbf{i}_t))\|_2\|\mathbf{W}_{hi}\|_2$$

$$+ \left\|\frac{\partial \xi_m}{\partial \mathbf{c}_t}\right\|\|\text{Diag}(\mathbf{c}_{t-1} \odot \sigma'(\mathbf{f}_t))\|_2\|\mathbf{W}_{hf}\|_2$$

$$+ \left\|\frac{\partial \xi_m}{\partial \mathbf{c}_t}\right\|\|\text{Diag}(\sigma(\mathbf{i}_t) \odot \tanh'(\mathbf{a}_t))\|_2\|\mathbf{W}_{ha}\|_2$$

$$+ \left\|\frac{\partial \xi_m}{\partial \mathbf{h}_t}\right\|\|\text{Diag}(\tanh(\mathbf{c}_t) \odot \sigma'(\mathbf{o}_t))\|_2\|\mathbf{W}_{ho}\|_2.$$

Using the property of norms, (8) and (17), we have

$$\left\|\frac{\partial \xi_m}{\partial \mathbf{h}_{t-1}}\right\|^2 \leq 4\left(\left\|\frac{\partial \xi_m}{\partial \mathbf{c}_t}\right\|^2\frac{1}{16}\|\mathbf{W}_{hi}\|_2^2 + \left\|\frac{\partial \xi_m}{\partial \mathbf{c}_t}\right\|^2\frac{1}{16}\|\text{Diag}(\mathbf{c}_{t-1})\|_2^2\|\mathbf{W}_{hf}\|_2^2\right.$$

$$\left. + \left\|\frac{\partial \xi_m}{\partial \mathbf{c}_t}\right\|^2\|\mathbf{W}_{ha}\|_2^2 + \left\|\frac{\partial \xi_m}{\partial \mathbf{h}_t}\right\|^2\frac{1}{16}\|\mathbf{W}_{ho}\|_2^2\right)$$

$$\leq \left(\frac{1}{2}\|\mathbf{W}_{hi}\|_2^2 + \frac{1}{2}\|\mathbf{W}_{hf}\|_2^2\|\text{Diag}(\mathbf{c}_{t-1})\|_2^2 + 8\|\mathbf{W}_{ha}\|_2^2 + \frac{1}{4}\|\mathbf{W}_{ho}\|_2^2\right)\left\|\frac{\partial \xi_m}{\partial \mathbf{h}_t}\right\|^2$$

$$\left(\frac{1}{2}\|\mathbf{W}_{hi}\|_2^2 + \frac{1}{2}\|\mathbf{W}_{hf}\|_2^2\|\text{Diag}(\mathbf{c}_{t-1})\|_2^2 + 8\|\mathbf{W}_{ha}\|_2^2\right)\left\|\frac{\partial \xi_m}{\partial \mathbf{c}_{t+1}}\right\|^2.$$

Summing over all the samples in the same minibatch, we have

$$\sum_{k=1}^N\left\|\frac{\partial \xi_m}{\partial \mathbf{h}_{t-1}^k}\right\|^2 \leq \left(\frac{1}{2}\|\mathbf{W}_{hi}\|_2^2 + \frac{\gamma_2^2}{2}\|\mathbf{W}_{hf}\|_2^2 + 8\|\mathbf{W}_{ha}\|_2^2 + \frac{1}{2}\|\mathbf{W}_{ho}\|_2^2\right)\sum_{k=1}^N\left\|\frac{\partial \xi_m}{\partial \mathbf{h}_t^k}\right\|^2$$

$$+ \left(\frac{1}{2}\|\mathbf{W}_{hi}\|_2^2 + \frac{\gamma_2^2}{2}\|\mathbf{W}_{hf}\|_2^2 + 8\|\mathbf{W}_{ha}\|_2^2\right)\sum_{k=1}^N\left\|\frac{\partial \xi_m}{\partial \mathbf{c}_{t+1}^k}\right\|^2.$$

### F.5 Proof for Proposition 3.4

Similar to the derivation in the proof for Propositions 3.1, we have

$$\frac{\partial \xi_m}{\partial \mathbf{c}_t^k} = \frac{\partial \xi_m}{\partial \mathbf{h}_t^k}\mathrm{Diag}(\sigma(\tilde{\mathbf{o}}_t) \odot \tanh'(\mathbf{c}_t^k)) + \frac{\partial \xi_m}{\partial \mathbf{c}_{t+1}}\mathrm{Diag}(\sigma(\tilde{\mathbf{f}}_{t+1}^k)).$$

Thus

$$\left\| \frac{\partial \xi_m}{\partial \mathbf{c}_t^k} \right\|^2 \le \left( \left\| \frac{\partial \xi_m}{\partial \mathbf{h}_t^k} \right\| + \left\| \frac{\partial \xi_m}{\partial \mathbf{c}_{t+1}^k} \right\| \right)^2 \le 2 \left\| \frac{\partial \xi_m}{\partial \mathbf{h}_t^k} \right\|^2 + 2 \left\| \frac{\partial \xi_m}{\partial \mathbf{c}_{t+1}^k} \right\|^2.$$

To bound the relationship between $\sum_{k=1}^{N} \left\| \frac{\partial \xi_m}{\partial \mathbf{h}_{t-1}^k} \right\|^2$ and $(\sum_{k=1}^{N} \left\| \frac{\partial \xi_m}{\partial \mathbf{h}_t^k} \right\|^2, \sum_{k=1}^{N} \left\| \frac{\partial \xi_m}{\partial \mathbf{c}_{t+1}^k} \right\|^2)$, we split the process in the following three steps.

1. Relate $\sum_{k=1}^{N} \left\| \frac{\partial \xi_m}{\partial \mathbf{h}_{t-1}^k} \right\|^2$ with $\sum_{k=1}^{N} \left\| \frac{\partial \xi_m}{\partial \mathbf{i}_t^k} \right\|^2$, $\sum_{k=1}^{N} \left\| \frac{\partial \xi_m}{\partial \mathbf{f}_t^k} \right\|^2$, $\sum_{k=1}^{N} \left\| \frac{\partial \xi_m}{\partial \mathbf{a}_t^k} \right\|^2$ and $\sum_{k=1}^{N} \left\| \frac{\partial \xi_m}{\partial \mathbf{o}_t^k} \right\|^2$;

2. Relate $\sum_{k=1}^{N} \left\| \frac{\partial \xi_m}{\partial \mathbf{i}_t^k} \right\|^2$, $\sum_{k=1}^{N} \left\| \frac{\partial \xi_m}{\partial \mathbf{f}_t^k} \right\|^2$, $\sum_{k=1}^{N} \left\| \frac{\partial \xi_m}{\partial \mathbf{a}_t^k} \right\|^2$, $\sum_{k=1}^{N} \left\| \frac{\partial \xi_m}{\partial \mathbf{o}_t^k} \right\|^2$ with $\sum_{k=1}^{N} \left\| \frac{\partial \xi_m}{\partial \tilde{\mathbf{i}}_t^k} \right\|^2$, $\sum_{k=1}^{N} \left\| \frac{\partial \xi_m}{\partial \tilde{\mathbf{f}}_t^k} \right\|^2$, $\sum_{k=1}^{N} \left\| \frac{\partial \xi_m}{\partial \tilde{\mathbf{a}}_t^k} \right\|^2$, $\sum_{k=1}^{N} \left\| \frac{\partial \xi_m}{\partial \tilde{\mathbf{o}}_t^k} \right\|^2$;

3. Relate $\sum_{k=1}^{N} \left\| \frac{\partial \xi_m}{\partial \tilde{\mathbf{i}}_t^k} \right\|^2$, $\sum_{k=1}^{N} \left\| \frac{\partial \xi_m}{\partial \tilde{\mathbf{f}}_t^k} \right\|^2$, $\sum_{k=1}^{N} \left\| \frac{\partial \xi_m}{\partial \tilde{\mathbf{a}}_t^k} \right\|^2$, $\sum_{k=1}^{N} \left\| \frac{\partial \xi_m}{\partial \tilde{\mathbf{o}}_t^k} \right\|^2$ with $\sum_{k=1}^{N} \left\| \frac{\partial \xi_m}{\partial \mathbf{h}_t^k} \right\|^2$ and $\sum_{k=1}^{N} \left\| \frac{\partial \xi_m}{\partial \mathbf{c}_{t+1}^k} \right\|^2$.

**Step 1.** By chain rule, we have

$$\begin{aligned}
\frac{\partial \xi_m}{\partial \mathbf{h}_{t-1}^k} &= \frac{\partial \xi_m}{\partial \mathbf{i}_t^k}\frac{\partial \mathbf{i}_t^k}{\partial \mathbf{h}_{t-1}^k} + \frac{\partial \xi_m}{\partial \mathbf{f}_t^k}\frac{\partial \mathbf{f}_t^k}{\partial \mathbf{h}_{t-1}^k} + \frac{\partial \xi_m}{\partial \mathbf{a}_t^k}\frac{\partial \mathbf{a}_t^k}{\partial \mathbf{h}_{t-1}^k} + \frac{\partial \xi_m}{\partial \mathbf{o}_t^k}\frac{\partial \mathbf{o}_t^k}{\partial \mathbf{h}_{t-1}^k} \\
&= \frac{\partial \xi_m}{\partial \mathbf{i}_t^k}\mathbf{W}_{hi} + \frac{\partial \xi_m}{\partial \mathbf{f}_t^k}\mathbf{W}_{hf} + \frac{\partial \xi_m}{\partial \mathbf{a}_t^k}\mathbf{W}_{ha} + \frac{\partial \xi_m}{\partial \mathbf{o}_t^k}\mathbf{W}_{ho}.
\end{aligned}$$

Thus using the property of norms, we have

$$\begin{aligned}
\sum_{k=1}^{N} \left\| \frac{\partial \xi_m}{\partial \mathbf{h}_{t-1}^k} \right\|^2 &= \sum_{k=1}^{N} \left\| \frac{\partial \xi_m}{\partial \mathbf{i}_t^k}\mathbf{W}_{hi} + \frac{\partial \xi_m}{\partial \mathbf{f}_t^k}\mathbf{W}_{hf} + \frac{\partial \xi_m}{\partial \mathbf{a}_t^k}\mathbf{W}_{ha} + \frac{\partial \xi_m}{\partial \mathbf{o}_t^k}\mathbf{W}_{ho} \right\|^2 \\
&\le \sum_{k=1}^{N} 4 \left( \left\| \frac{\partial \xi_m}{\partial \mathbf{i}_t^k} \right\|^2 \|\mathbf{W}_{hi}\|_2^2 + \left\| \frac{\partial \xi_m}{\partial \mathbf{f}_t^k} \right\|^2 \|\mathbf{W}_{hf}\|_2^2 \right. \\
&\qquad\qquad \left. + \left\| \frac{\partial \xi_m}{\partial \mathbf{a}_t^k} \right\|^2 \|\mathbf{W}_{ha}\|_2^2 + \left\| \frac{\partial \xi_m}{\partial \mathbf{o}_t^k} \right\|^2 \|\mathbf{W}_{ho}\|_2^2 \right) \\
&= 4 \|\mathbf{W}_{hi}\|_2^2 \sum_{k=1}^{N} \left\| \frac{\partial \xi_m}{\partial \mathbf{i}_t^k} \right\|^2 + 4 \|\mathbf{W}_{hf}\|_2^2 \sum_{k=1}^{N} \left\| \frac{\partial \xi_m}{\partial \mathbf{f}_t^k} \right\|^2 \\
&\quad + 4 \|\mathbf{W}_{ha}\|_2^2 \sum_{k=1}^{N} \left\| \frac{\partial \xi_m}{\partial \mathbf{a}_t^k} \right\|^2 + 4 \|\mathbf{W}_{ho}\|_2^2 \sum_{k=1}^{N} \left\| \frac{\partial \xi_m}{\partial \mathbf{o}_t^k} \right\|^2.
\end{aligned}$$

**Step 2.** In the following, we first derive the Jacobian of the output after batch normalization w.r.t. the input before batch normalization. Denote $\mathbf{I}_t = [\mathbf{i}_t^1, \mathbf{i}_t^2, \cdots, \mathbf{i}_t^N] \in \mathbb{R}^{N \times d}$ and $\tilde{\mathbf{I}}_t = [\tilde{\mathbf{i}}_t^1, \tilde{\mathbf{i}}_t^2, \cdots, \tilde{\mathbf{i}}_t^N] \in \mathbb{R}^{N \times d}$ where $\tilde{\mathbf{i}}_t^k = \mathcal{BN}(\mathbf{i}_t^k), k \in \{1, \cdots, N\}$, the pre-activation input gate values before and after batch normalization, respectively. Similarly, $\mathbf{F}, \mathbf{A}, \mathbf{O} \in \mathbb{R}^{N \times d}$ and $\tilde{\mathbf{F}}, \tilde{\mathbf{A}}, \tilde{\mathbf{O}} \in \mathbb{R}^{N \times d}$ are the pre-activation before and after batch normalization for the forget gates, cell update and output gates,

respectively. From Theorem 4.1 in [25], we have

$$\left\|\frac{\partial\xi_m}{\mathbf{I}_{:,j}}\right\|^2 \le \frac{g_i^2}{\sigma_i^2}\left(\left\|\frac{\partial\xi_m}{\tilde{\mathbf{I}}_{:,j}}\right\|^2 - \frac{1}{N}\langle\mathbf{1}, \frac{\partial\xi_m}{\partial\tilde{\mathbf{I}}_{:,j}}\rangle^2 - \frac{1}{\sqrt{N}}\langle\frac{\partial\xi_m}{\partial\tilde{\mathbf{I}}_{:,j}}, \mathbf{Z}_{:,j}\rangle^2\right) \le \frac{g_i^2}{\sigma_i^2}\left\|\frac{\partial\xi_m}{\tilde{\mathbf{I}}_{:,j}}\right\|^2,$$

where $\mathbf{Z}_{:,j}$ is the z-normalized vector with zero-mean and unit variance in the $j$th dimension. Similarly:

$$\left\|\frac{\partial\xi_m}{\mathbf{F}_{:,j}}\right\|^2 \le \frac{g_f^2}{\sigma_f^2}\left\|\frac{\partial\xi_m}{\tilde{\mathbf{F}}_{:,j}}\right\|^2, \quad \left\|\frac{\partial\xi_m}{\mathbf{A}_{:,j}}\right\|^2 \le \frac{g_a^2}{\sigma_a^2}\left\|\frac{\partial\xi_m}{\tilde{\mathbf{A}}_{:,j}}\right\|^2, \quad \left\|\frac{\partial\xi_m}{\mathbf{O}_{:,j}}\right\|^2 \le \frac{g_o^2}{\sigma_o^2}\left\|\frac{\partial\xi_m}{\tilde{\mathbf{O}}_{:,j}}\right\|^2.$$

Combining with the results from Step 1, we have

$$\begin{aligned}
\sum_{k=1}^{N}\left\|\frac{\partial\xi_m}{\partial\mathbf{h}_{t-1}^k}\right\|^2 &\le 4\|\mathbf{W}_{hi}\|_2^2\sum_{k=1}^{N}\left\|\frac{\partial\xi_m}{\partial\mathbf{i}_t^k}\right\|^2 + 4\|\mathbf{W}_{hf}\|_2^2\sum_{k=1}^{N}\left\|\frac{\partial\xi_m}{\partial\mathbf{f}_t^k}\right\|^2 \\
&\quad + 4\|\mathbf{W}_{ha}\|_2^2\sum_{k=1}^{N}\left\|\frac{\partial\xi_m}{\partial\mathbf{a}_t^k}\right\|^2 + 4\|\mathbf{W}_{ho}\|_2^2\sum_{k=1}^{N}\left\|\frac{\partial\xi_m}{\partial\mathbf{o}_t^k}\right\|^2 \\
&\le 4\|\mathbf{W}_{hi}\|_2^2\sum_{j=1}^{d}\left\|\frac{\partial\xi_m}{\partial\mathbf{I}_{:,j}}\right\|^2 + 4\|\mathbf{W}_{hf}\|_2^2\sum_{j=1}^{d}\left\|\frac{\partial\xi_m}{\partial\mathbf{F}_{:,j}}\right\|^2 \\
&\quad + 4\|\mathbf{W}_{ha}\|_2^2\sum_{j=1}^{d}\left\|\frac{\partial\xi_m}{\partial\mathbf{A}_{:,j}}\right\|^2 + 4\|\mathbf{W}_{ho}\|_2^2\sum_{j=1}^{d}\left\|\frac{\partial\xi_m}{\partial\mathbf{O}_{:,j}}\right\|^2 \\
&\le 4\|\mathbf{W}_{hi}\|_2^2\frac{g_i^2}{\sigma_i^2}\sum_{j=1}^{d}\left\|\frac{\partial\xi_m}{\partial\tilde{\mathbf{I}}_{:,j}}\right\|^2 + 4\|\mathbf{W}_{hf}\|_2^2\frac{g_f^2}{\sigma_f^2}\sum_{j=1}^{d}\left\|\frac{\partial\xi_m}{\partial\tilde{\mathbf{F}}_{:,j}}\right\|^2 \\
&\quad + 4\|\mathbf{W}_{ha}\|_2^2\frac{g_a^2}{\sigma_a^2}\sum_{j=1}^{d}\left\|\frac{\partial\xi_m}{\partial\tilde{\mathbf{A}}_{:,j}}\right\|^2 + 4\|\mathbf{W}_{ho}\|_2^2\frac{g_o^2}{\sigma_o^2}\sum_{j=1}^{d}\left\|\frac{\partial\xi_m}{\partial\tilde{\mathbf{O}}_{:,j}}\right\|^2 \\
&\le 4\|\mathbf{W}_{hi}\|_2^2\frac{g_i^2}{\sigma_i^2}\sum_{k=1}^{N}\left\|\frac{\partial\xi_m}{\partial\tilde{\mathbf{i}}_t^k}\right\|^2 + 4\|\mathbf{W}_{hf}\|_2^2\frac{g_f^2}{\sigma_f^2}\sum_{k=1}^{N}\left\|\frac{\partial\xi_m}{\partial\tilde{\mathbf{f}}_t^k}\right\|^2 \\
&\quad + 4\|\mathbf{W}_{ha}\|_2^2\frac{g_a^2}{\sigma_a^2}\sum_{k=1}^{N}\left\|\frac{\partial\xi_m}{\partial\tilde{\mathbf{a}}_t^k}\right\|^2 + 4\|\mathbf{W}_{ho}\|_2^2\frac{g_o^2}{\sigma_o^2}\sum_{k=1}^{N}\left\|\frac{\partial\xi_m}{\partial\tilde{\mathbf{o}}_t^k}\right\|^2.
\end{aligned}$$

**Step 3.** By chain rule, we have

$$\frac{\partial\xi_m}{\partial\tilde{\mathbf{i}}_t^k} = \frac{\partial\xi_m}{\partial\mathbf{c}_t^k}\text{Diag}(\tanh(\tilde{\mathbf{a}}_t^k)\odot\sigma'(\tilde{\mathbf{i}}_t^k)), \quad \frac{\partial\xi_m}{\partial\tilde{\mathbf{f}}_t^k} = \frac{\partial\xi_m}{\partial\mathbf{c}_t^k}\text{Diag}(\mathbf{c}_{t-1}^k\odot\sigma'(\tilde{\mathbf{f}}_t^k)),$$

$$\frac{\partial\xi_m}{\partial\tilde{\mathbf{a}}_t^k} = \frac{\partial\xi_m}{\partial\mathbf{c}_t^k}\text{Diag}(\sigma(\tilde{\mathbf{i}}_t^k)\odot\tanh'(\tilde{\mathbf{a}}_t^k)), \quad \frac{\partial\xi_m}{\partial\tilde{\mathbf{o}}_t^k} = \frac{\partial\xi_m}{\partial\mathbf{h}_t^k}\text{Diag}(\tanh(\mathbf{c}_t^k)\odot\sigma'(\tilde{\mathbf{o}}_t^k)).$$

Using the property of norms, and the bound on sigmoid, tanh and their derivatives in (8), we have

$$\left\|\frac{\partial\xi_m}{\partial\tilde{\mathbf{i}}_t^k}\right\|^2 \le \frac{1}{16}\left\|\frac{\partial\xi_m}{\partial\mathbf{c}_t^k}\right\|^2, \quad \left\|\frac{\partial\xi_m}{\partial\tilde{\mathbf{f}}_t^k}\right\|^2 \le \frac{1}{16}\left\|\text{Diag}(\mathbf{c}_{t-1}^k)\right\|_2^2\left\|\frac{\partial\xi_m}{\partial\mathbf{c}_t^k}\right\|^2,$$

$$\left\|\frac{\partial\xi_m}{\partial\tilde{\mathbf{a}}_t^k}\right\|^2 \le \left\|\frac{\partial\xi_m}{\partial\mathbf{c}_t^k}\right\|^2, \quad \left\|\frac{\partial\xi_m}{\partial\tilde{\mathbf{o}}_t^k}\right\|^2 \le \frac{1}{16}\left\|\frac{\partial\xi_m}{\partial\mathbf{h}_t^k}\right\|^2.$$

From the results in Step 2, summing over all the samples in the same minibatch, we have

$$\sum_{k=1}^{N} \left\| \frac{\partial \xi_m}{\partial \mathbf{h}_{t-1}^k} \right\|^2 \leq 4 \left\| \mathbf{W}_{hi} \right\|_2^2 \frac{g_i^2}{\sigma_i^2} \sum_{k=1}^{N} \left\| \frac{\partial \xi_m}{\partial \tilde{\mathbf{i}}_t^k} \right\|^2 + 4 \left\| \mathbf{W}_{hf} \right\|_2^2 \frac{g_f^2}{\sigma_f^2} \sum_{k=1}^{N} \left\| \frac{\partial \xi_m}{\partial \tilde{\mathbf{f}}_t^k} \right\|^2 \left\| \mathrm{Diag}(\mathbf{c}_{t-1}^k) \right\|^2$$

$$+ 4 \left\| \mathbf{W}_{ha} \right\|_2^2 \frac{g_a^2}{\sigma_a^2} \sum_{k=1}^{N} \left\| \frac{\partial \xi_m}{\partial \tilde{\mathbf{a}}_t^k} \right\|^2 + 4 \left\| \mathbf{W}_{ho} \right\|_2^2 \frac{g_o^2}{\sigma_o^2} \sum_{k=1}^{N} \left\| \frac{\partial \xi_m}{\partial \tilde{\mathbf{o}}_t^k} \right\|^2$$

$$\leq \left\| \mathbf{W}_{hi} \right\|_2^2 \frac{g_i^2}{\sigma_i^2} \sum_{k=1}^{N} \frac{1}{4} \left\| \frac{\partial \xi_m}{\partial \mathbf{c}_t^k} \right\|^2 + \left\| \mathbf{W}_{hf} \right\|_2^2 \frac{g_f^2}{\sigma_f^2} \max_{1 \leq k \leq N} \left\| \mathrm{Diag}(\mathbf{c}_{t-1}^k) \right\|_2^2 \sum_{k=1}^{N} \frac{1}{4} \left\| \frac{\partial \xi_m}{\partial \mathbf{c}_t^k} \right\|^2$$

$$+ \left\| \mathbf{W}_{ha} \right\|_2^2 \frac{g_a^2}{\sigma_a^2} \sum_{k=1}^{N} 4 \left\| \frac{\partial \xi_m}{\partial \mathbf{c}_t^k} \right\|^2 + \left\| \mathbf{W}_{ho} \right\|_2^2 \frac{g_o^2}{\sigma_o^2} \sum_{k=1}^{N} \frac{1}{4} \left\| \frac{\partial \xi_m}{\partial \mathbf{h}_t^k} \right\|^2 .$$

$$\sum_{k=1}^{N} \left\| \frac{\partial \xi_m}{\partial \mathbf{h}_{t-1}^k} \right\|^2 \leq \left( \frac{1}{2} \frac{g_i^2}{\sigma_i^2} \left\| \mathbf{W}_{hi} \right\|_2^2 + \frac{1}{2} \gamma_2^2 \frac{2 g_f^2}{\sigma_f^2} \left\| \mathbf{W}_{hf} \right\|_2^2 + 8 \frac{g_a^2}{\sigma_a^2} \left\| \mathbf{W}_{ha} \right\|_2^2 + \frac{1}{4} \frac{g_o^2}{\sigma_o^2} \left\| \mathbf{W}_{ho} \right\|_2^2 \right) \sum_{k=1}^{N} \left\| \frac{\partial \xi_m}{\partial \mathbf{h}_t^k} \right\|^2$$

$$+ \left( \frac{1}{2} \frac{g_i^2}{\sigma_i^2} \left\| \mathbf{W}_{hi} \right\|_2^2 + \frac{1}{2} \gamma_2^2 \frac{2 g_f^2}{\sigma_f^2} \left\| \mathbf{W}_{hf} \right\|_2^2 + 8 \frac{g_a^2}{\sigma_a^2} \left\| \mathbf{W}_{ha} \right\|_2^2 \right) \sum_{k=1}^{N} \left\| \frac{\partial \xi_m}{\partial \mathbf{c}_{t+1}^k} \right\|^2 .$$