[Reviews · NeurIPS 2019]

Reviewer 1



I think building the connection between the norm of weight matrices and gradient is original, but using normalization to improve the accuracy loss caused by quantization is not so impressive. My detailed concerns are listed as follows: 1. In Eq. (4), the condition of gradient exploding is lambda_1>1. Although Table 1 shows the matrix norm, it is not a direct way. Can you clearly visualize the lambda_1 values in several typical LSTMs, before and after quantization without normalization? 2. In the three normalizations, the trainable scaling factors (g) will still affect the value of lambda_1 (thus affects the exploding condition), but the authors mentioned little on this. So can you present the g and lambda_1 values in typical LSTMs before and after quantization with normalization? 3. In sequential MNIST task, the batch normalization (shared) method totally failed. Could you explain why the behavior on this task is such different from others? 4. I noted that the quantization with scaling factors (e.g. BWN, TWN, alternating LSTM) can also improve the accuracy. But I did not see these results in Table 4. Could you please provide them for better comparison? 5. Do your conclusions still hold in GRU? You can also conduct the similar chain rule analysis during back propagation. 6. How about in multilayer LSTMs?

Reviewer 2



While in recent years a number of extreme low-precision quantization techniques were developed for DNNs, they were not directly applicable to recurrent architectures. In recent work [1] an extreme low-precision quantization method was proposed that utilizes batch normalization and compresses recurrent neural networks without a large drop in accuracy achieving state-of-the-art performance. In this paper, the authors proposed a theoretical explanation of the difficulties of training LSTMs with low-precision weights and practically explored a combination of different normalization techniques with different quantization schemes. The authors experimentally showed that simple introduction of weight or layer normalization allows applying many standard quantization techniques without modifications. Comments, suggestions, and questions: Figure 1 is quite difficult to read. I would suggest using a logarithmic scale. I would also suggest adding a reference to footnote 3 in the caption of the figure in order to increase clarity Line 183 “On text8, we use a” -> “On Text8, we use a” Do I understand correctly that simple application of batch normalization works better or similar to SBN? In [1] the authors claim that the size of their network is 5 KBytes, while in the Table4, the SBN size is 5526KBytes? Can the authors please clarify? Overall, the propositions on upper bounds are quite straight-forward. The experimental results are simple combinations of previous works. Nonetheless, it is quite interesting that the simple usage of different normalization techniques makes different quantization schemes applicable again. [1] A. Ardakani, Z. Ji, S. C. Smithson, B. H. Meyer, and W. J. Gross. Learning recurrent binary/ternary weights. In International Conference on Learning Representations, 2019. I would like to thanks the authors for their comments. I updated my score since the answers to my questions are mostly positive.

Reviewer 3



The strengths of the paper besides the above contributions: - The paper is clearly written. - The systematic comparisons are thorough and convincing. - The theoretic analysis is useful for understanding the problem and provides the justification for the solutions. - Code is provided for reproducibility. The weaknesses that could be further improved: - The approach used in the theoretic analysis in the paper isn't able to give a necessary or a sufficient condition for the exploding gradient problem for LSTM since lambda_2 in Proposition 2.1 is typically not zero. The analyses of the various normalization approaches are also only made on the upper bound of the gradient magnitude, but on the gradient itself. It would be great to have a more direct proof for the effectiveness of the solutions. - In the analysis in Section 3, it only studies that the increased scaling of the weight matrices does not affect the upper bound of the gradient magnitude due to \sigma. It would be good to also analyze the effect of all the normalization scaling parameters 'g' w.r.t. the exploding gradient problem. - It would make the paper even stronger if it can show the normalization techniques still make 1-bit or 2-bit quantized LSTM achieve comparable performance to a full-precision counterpart on a much larger network consisting of multiple LSTM layers, e.g. for machine translation or speech recognition, where there is a strong practical need for compressing the models especially for the embedded/mobile use case.

[Author Response · NeurIPS 2019]

<span style="color:blue">Reviewer 1</span> **"using normalization to improve the accuracy loss caused by quantization is not so impressive"**: Our
main contribution is on the theoretical analysis of quantized LSTM training and normalization. On the empirical results,
Tables 2-4 show that accuracy loss due to quantization can be significant (especially for BinaryConnect and TerConnect),
and normalization helps to recover performance, sometimes to the level comparable with the full-precision baseline.

**"$\lambda_1$ values ... before and after quantization without normalization"**: Figure(a) shows $\lambda_1$ values for full-precision
LSTM, binarized (using BinaryConnect (BC)), and ternarized (using TerConnect (TC)) LSTMs without normalization,
on character language modeling with *Penn Treebank*. As can be seen, after BC and TC, $\lambda_1$ is much larger.

**"$g$ and $\lambda_1$ values ... before and after quantization with normalization"**: Figure(a) shows $\lambda_1$ values for the binarized
LSTM with weight/layer normalization. Figure(b) shows $\lambda_1$ values with batch normalization.[1] As can be seen,
normalization makes $\lambda_1$ in binarized LSTM much smaller. Finally, Figure(c) shows the corresponding $g$ values.
Because of lack of space, results for the ternarized LSTM are not shown.

| precision | quantization | normalization | MNIST | pMNIST | size |
|---|---|---|---|---|---|
| full | - | - | 98.9 | 90.2 | 159 |
| 1-bit | BWN | - | 98.6 | 89.7 | 8 |
| | | weight | 98.7 | **91.3** | 11 |
| | | layer | **98.8** | 90.8 | 14 |
| | | batch (shared) | 20.6 | 40.1 | 21 |
| | | batch (separate) | 98.6 | 91.1 | 4914 |
| 2-bit | TWN | - | 98.6 | 90.4 | 13 |
| | | weight | 98.6 | 92.1 | 16 |
| | | layer | **98.8** | 91.7 | 18 |
| | | batch (shared)) | 26.5 | 38.3 | 25 |
| | | batch (separate) | 98.7 | **93.1** | 4919 |

(a) $\lambda_1$ in Propositions 2.1, 3.1-3.2.   (b) $\lambda_1$ in Propositions 3.3-3.4.   (c) $g$.

**"sequential MNIST task, the batch normalization (shared) method totally failed"**: In this task, each time step
corresponds to an input pixel. The use of shared batch normalization statistics implicitly assumes different pixels have
similar characteristics. However, this may not be reasonable (e.g., pixels around the edge are typically darker).

**" quantization with scaling factors"**: The table above adds BWN/TWN results on sequential MNIST task (Table 4).
The weight/layer normalized quantized LSTMs have comparable results as full-precision baselines, but much smaller.

**"conclusions still hold in GRU?"**: This is an open issue, as analysis for GRU is different. We leave this as future work.

**"How about in multilayer LSTMs?"**: Below we add results on 2-layer LSTM for *Penn Treebank* task (first number:
test PPL, second: size(KB), "no" means no normalization). Again, BinaryConnect/TerConnect fail when directly used,
but achieve results comparable as the full-precision baseline when normalized, while being much smaller in size.
**Full-precision:** no(81.67, 26427) weight(81.82, 26467) layer(81.02, 26508) batch-shared(80.60,26589) batch-separate(81.42,29352)
**BinaryConnect:** no(134.47, 846) weight(81.86, 886) layer(82.10, 927) batch-shared(80.55, 1008) batch-separate(81.48, 3771)
**TerConnect:** no(639.61,1671) weight(80.54, 1711) layer(80.06, 1752) batch-shared(79.25, 1833) batch-separate(79.89, 4596)

<span style="color:blue">Reviewer 2</span> **"simple application of batch normalization works better or similar to SBN?"**: Yes.

**"In [1] the authors claim that the size of their network is 5 KBytes, while in the Table4, the SBN size is
5526KBytes?"**: As mentioned in line 185-188, [1] does not count the additional storage for the full-precision mean
and standard deviation statistics (in batch normalization) at the $T$ time steps. For the sequential MNIST task, $T = 784$,
and these statistics are in fact even much larger than the model itself. The unnormalized binary model in our Table 4
has size 8 KBytes. This is slightly larger than 5 KBytes because we do not quantize $\mathbf{W}_{x*}$ (Appendix A.3). For detailed
model size analysis, please see Remark 3.2.

**"propositions on upper bounds are quite straight-forward"**: LSTM, due to introduction of $c_t$, is more difficult to
analyze than vanilla RNN. Adding normalization makes the analysis even more non-trivial. For example, in batch
normalization, samples in a minibatch become related and the gradient flow is intertwined in different dimensions
(please see steps 1-3 in Appendix D.8). Also, we are the first to derive these upper bounds to analyze why quantized
LSTM is difficult to train, and how normalization helps.

**"experimental results are simple combinations of previous works"**: In the experiments, we thoroughly run and
study various 1-bit and 2-bit quantization methods with different normalizations (weight/layer/batch) on different tasks.

<span style="color:blue">Reviewer 3</span> **"$\lambda_2$ typically not zero"**: We expect a nonzero $\lambda_2$ will make gradient explosion happen more easily (lines
90-91).

**"only made on the upper bound of the gradient magnitude"**: We agree that a lower bound will also be useful.
However, even for vanilla RNN, only an upper bound can be derived in (Pascanu et al., 2012).

**"effect of all the normalization scaling parameters 'g'"**: Please refer to our third reply to Reviewer 1.

**"larger network consisting of multiple LSTM layers"** : Please refer to our last reply to Reviewer 1.

$\sum_{k=1}^{N} \| \frac{\partial \xi_m}{\partial \mathbf{h}_t^k} \|^2$. With an abuse of notation, we still call this value $\lambda_1$.

## Footnotes

[1]The bounds in Propositions 3.3 and 3.4 are based on the squared weight norm. Hence, Figure(b) plots the coefficient before


[Meta-Review · NeurIPS 2019]

The paper provides a better understanding of the task of quantizing LSTMs. This understanding is translated to simpler methods essentially involving standard normalizations combined with standard quantization methods. On the one hand, the reviewers feel the analysis is not very complicated and the conclusion that normalization helps quantization is not surprising. On the other hand, the paper does provide a convincing analysis resulting in a simple and effective solution improving the results of previous papers. Despite not having any major breakthrough, the paper is likely to be interesting for the NeurIPS audience. It is borderline, but I tend to recommend accepting it, pending the quality of other submitted papers.